

# Generalized charges, part I:
# Invertible symmetries and higher representations

**Lakshya Bhardwaj and Sakura Schäfer-Nameki**

Mathematical Institute, University of Oxford, Andrew-Wiles Building,
Woodstock Road, Oxford, OX2 6GG, UK

## Abstract

$q$-charges describe the possible actions of a generalized symmetry on $q$-dimensional operators. In Part I of this series of papers, we describe $q$-charges for invertible symmetries; while the discussion of $q$-charges for non-invertible symmetries is the topic of Part II. We argue that $q$-charges of a standard global symmetry, also known as a 0-form symmetry, correspond to the so-called $(q+1)$-representations of the 0-form symmetry group, which are natural higher-categorical generalizations of the standard notion of representations of a group. This generalizes already our understanding of possible charges under a 0-form symmetry! Just like local operators form representations of the 0-form symmetry group, higher-dimensional extended operators form higher-representations. This statement has a straightforward generalization to other invertible symmetries: $q$-charges of higher-form and higher-group symmetries are $(q+1)$-representations of the corresponding higher-groups. There is a natural extension to higher-charges of non-genuine operators (i.e. operators that are attached to higher-dimensional operators), which will be shown to be intertwiners of higher-representations. This brings into play the higher-categorical structure of higher-representations. We also discuss higher-charges of twisted sector operators (i.e. operators that appear at the boundary of topological operators of one dimension higher), including operators that appear at the boundary of condensation defects.

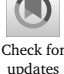

# 1   Introduction and summary of results

## 1.1   Introduction

The recent developments on non-invertible symmetries hold promising potential to open an exciting chapter in the study of non-perturbative phenomena in quantum field theory (QFT). To study systems with conventional group-like symmetries, representation theory is of course indispensable, as it describes the action of these symmetries on physical states and local operators. Likewise, we will argue that the key to unlocking the full utility of generalized, in particular non-invertible, symmetries is to understand their action on local and extended operators of various dimensions. Said differently, the key is to determine the **generalized charges** carried by operators in a QFT with generalized global symmetries. This will be laid out in this series of papers, where the present one is the first, with subsequent followups in Parts II [1] and III [2]. The role that representation theory plays for groups, is replaced here by higher-representations, which intimately tie into the categorical nature of the symmetries.

Although at this point in time firmly established as a central tool in theoretical physics, historically, group theory and representation theory of groups has faced an upwards battle. Eugene Wigner, who was one of the first to use group theory in the description of quantum mechanics [3], recalls that the advent of group theory in quantum mechanics was referred to by some as the "Gruppenpest" (German for "group plague"), a term allegedly coined by Schrödinger [4]. This sentiment was born out of the conviction that formal mathematics – in this case group theory – had no place in physics. Clearly history has proven Wigner and friends right, with group theory now a firmly established part of theoretical physics. Category theory has faced a similar battle in the past, justified or not. The case we would like to make here is that much like group theory is indispensable in describing physics, so is higher-category theory. In short we will make the case that what was group representation theory for physics in the 1920s, is higher-categories and the higher charges (as will be defined in this series of papers) for generalized symmetries 100 years later.

Higher-form symmetries [5] play an important role in these developments, and in this first paper we will show that already for these invertible (group-like) symmetries (including 0-form symmetries) there exist new, generalized charges. In particular, the standard paradigm of "$p$-dimensional operators are charged under $p$-form symmetries" turns out to only scratch the surface. We find that a $p$-form symmetry generically acts on defects of $q \geq p$ dimensions.

Naturally this leads then to the question, how the generalized categorical symmetries, including the non-invertible symmetries [6–61] act on operators. This will be the topic of part II [1] of this series.

In the following subsections, we begin with a summary and description of the results, in order to immediately share our excitement of these insights with the readers.

## 1.2 Generalized charges

The central topic of this series of papers is the development of generalized chages. In this paper, Part I, we discuss generalized charges for *invertible* generalized symmetries. This includes not only the standard group-like 0-form global symmetries, but also higher-form and higher-group symmetries. The extension to generalized charges of non-invertible generalized symmetries is the topic of discussion of the Part II in this series [1].

Before we begin, we will characterize generalized charges using the following terminology:

> **Definition**
>
> Generalized charges of $q$-dimensional operators are referred to as **$q$-charges**.

This definition is applicable to both invertible and non-invertible symmetries, and hence will be used throughout this series of papers. From this point onward, in this paper, we focus our attention to invertible symmetries.

Let us recall that some types of $q$-charges are well-understood [5]:

> **Statement 1.1: $p$-charges for $p$-form symmetries**
>
> $p$-charges of a $G^{(p)}$ $p$-form symmetry are representations of the group $G^{(p)}$.
>
> The most well-known case arises for $p = 0$, which states that 0-charges for $G^{(0)}$ 0-form symmetry are representations of the group $G^{(0)}$.

We argue in this paper that there is a natural extension of this fact to higher-charges: $q$-dimensional operators charged under $G^{(p)}$, i.e. $q$-charges, with $q > p$. The first extension

to consider is for $p = 0$, where higher-charges describe the action of 0-form symmetry on extended operators. Indeed, this case will be familiar, in that 0-form symmetries can act on extended operators by permuting them, as in the following example.

---

**Example 1.1: Higher-charges of charge conjugation 0-form symmetry**

Consider 4d pure Maxwell theory, which has a charge conjugation 0-form symmetry $G^{(0)} = \mathbb{Z}_2^{(0)}$. The theory also has a $G^{(1)} = U(1)_e^{(1)} \times U(1)_m^{(1)}$ 1-form symmetry, with $U(1)_e$ being the electric 1-form symmetry whose 1-charges are furnished by Wilson lines and $U(1)_m$ being the magnetic 1-form symmetry whose 1-charges are furnished by 't Hooft lines.

This theory furnishes 0-charges, 1-charges and 2-charges of the 0-form symmetry $G^{(0)} = \mathbb{Z}_2^{(0)}$:

1. **0-charges**: The charge conjugation acts on gauge field $A$ by changing its sign

$$A \longrightarrow -A, \tag{1}$$

which implies that the field strength $F(x)$, which is a local operator since it is gauge invariant, is acted upon by $G^{(0)} = \mathbb{Z}_2^{(0)}$

$$F(x) \longrightarrow -F(x), \tag{2}$$

thus furnishing a non-trivial 0-charge.

2. **1-charges**: A Wilson line of electric charge $e$ is exchanged with a Wilson line of charge $-e$ by the action of charge conjugation. Thus Wilson lines furnish non-trivial 1-charges of $G^{(0)} = \mathbb{Z}_2^{(0)}$.
   Similarly, a 't Hooft line of magnetic charge $m$ is exchanged with a 't Hooft line of charge $-m$ by the action of charge conjugation. Thus 't Hooft lines also furnish non-trivial 1-charges of $G^{(0)} = \mathbb{Z}_2^{(0)}$.

3. **2-charges**: Dually to the above actions of charge conjugation on 1-charges of 1-form symmetries, we have actions of charge conjugation on the topological surface operators

$$\begin{aligned} D_2^{(e,\alpha)} &= e^{i\alpha \int \star F}, \\ D_2^{(m,\alpha)} &= e^{i\alpha \int F}, \end{aligned} \tag{3}$$

generating the electric and magnetic 1-form symmetries

$$\begin{aligned} D_2^{(e,\alpha)} &\longleftrightarrow D_2^{(e,2\pi-\alpha)}, \\ D_2^{(m,\alpha)} &\longleftrightarrow D_2^{(m,2\pi-\alpha)}. \end{aligned} \tag{4}$$

Thus 1-form symmetry generators furnish non-trivial 2-charges of the 0-form symmetry $G^{(0)} = \mathbb{Z}_2^{(0)}$.

---

In the above example, the only non-trivial structure about the higher-charges is encoded in the $\mathbb{Z}_2$ exchange action. However, for a general 0-form symmetry the structure of higher-charges is pretty rich, and will be elucidated in depth in this paper. For now, we note that the general structure of $q$-form charges of 0-form symmetries is encapsulated in the following statement.

> **Statement 1.2: Generalized charges for 0-form symmetries**
>
> $q$-charges of a $G^{(0)}$ 0-form symmetry are **$(q + 1)$-representations** of the group $G^{(0)}$.

For $q = 0$, we obtain the well-known statement that 0-charges of a 0-form symmetry group $G^{(0)}$ are representations (also referred to as 1-representations) of $G^{(0)}$. However, for $q > 0$, this statement takes us into the subject of higher-representations, which are extremely natural higher-categorical generalizations of the usual (1-)representations. We will denote $(q + 1)$-representations by

$$\rho^{(q+1)}. \tag{5}$$

We describe the mathematical definition of higher-representations in appendix B. In the main text, instead of employing a mathematical approach, we will take a physical approach exploring the possible ways a 0-form symmetry group $G^{(0)}$ can act on $q$-dimensional operators. This naturally leads us to discover, that the physical concepts describing the action of $G^{(0)}$ on $q$-dimensional operators, correspond precisely to the mathematical structure of $(q + 1)$-representations of $G^{(0)}$. It is worthwhile emphasizing that this applies equally to finite but also continuous $G^{(0)}$: the definition of higher-representations and statement 1.2 is equally applicable for finite and for continuous 0-form symmetries.

   Naturally we should ask whether this extends to higher-form symmetries. Indeed, we find the following statement analogous to the statement 1.2:

> **Statement 1.3: Generalized charges for higher-form symmetries**
>
> $q$-charges of a $G^{(p)}$ $p$-form symmetry are $(q+1)$-representations of the associated $(p+1)$-group $\mathbb{G}^{(p+1)}_{G^{(p)}}$.

In order to explain this statement, we need to recall that a $(p+1)$-group $\mathbb{G}^{(p+1)}$ is a mathematical structure describing $r$-form symmetry groups for $0 \le r \le p$ along with possible interactions between the different $r$-form symmetry groups. Now, a $p$-form symmetry group $G^{(p)}$ naturally forms a $(p+1)$-group $\mathbb{G}^{(p+1)}_{G^{(p)}}$ whose component $r$-form symmetry groups are all trivial except for $r = p$. We will discuss the above statement 1.3 at length for $p = 1$-form symmetries in this paper. For the moment, let us note that the statement 1.1 is obtained as the special case $q = p$ of the above statement 1.3 because we have the identity

$$(p+1)\text{-representations of the }(p+1)\text{-group }\mathbb{G}^{(p+1)}_{G^{(p)}} = \text{Representations of the group }G^{(p)}. \tag{6}$$

As a final generalization in this direction, while remaining in the realm of invertible symmetries, we have the following general statement whose special cases are the previous two statements 1.2 and 1.3.

> **Statement 1.4: Generalized charges for higher-group symmetries**
>
> $q$-charges of a $\mathbb{G}^{(p)}$ $p$-group symmetry are $(q+1)$-representations of the $p$-group $\mathbb{G}^{(p)}$.

For higher-groups $(q + 1)$-representations will be denoted by

$$\boldsymbol{\rho}^{(q+1)}. \tag{7}$$

This covers all possible invertible generalized symmetries, taking into account interactions between different component $r$-form symmetry groups. We will discuss the above statement 1.4 at length for $p = 2$-group symmetries in this paper. As in the 0-form symmetry case, these statements apply equally to finite or to continuous higher-groups (and higher-form) symmetries.

## 1.3 Non-genuine generalized charges

The considerations of the above subsection are valid only for $q$-charges furnished by **genuine** $q$-dimensional operators. These are $q$-dimensional operators that exist on their own and do not need to be attached to any higher-dimensional operators in order to be well-defined. In this subsection, we discuss the $q$-charges that can be furnished by **non-genuine** $q$-dimensional operators which need to be attached to higher-dimensional operators in order to be well-defined.

---

**Example 1.2: Operators carrying gauge charges are non-genuine**

Examples of non-genuine operators are provided by operators which are not gauge invariant. Take the example of a $U(1)$ gauge theory with a scalar field $\phi$ of gauge charge $q$. An insertion $\phi(x)$ of the corresponding local operator is not gauge invariant and hence not a well-defined genuine local operator. However, one can obtain a gauge-invariant configuration

$$\phi(x)\exp\left( iq \int_x^\infty A \right), \tag{8}$$

by letting $\phi(x)$ lie at the end of a Wilson line operator of charge $q$. This configuration can be displayed diagrammatically as

$$\underset{W_q = e^{iq\int A}}{\rule{4cm}{0.4pt}\bullet}\ \overset{\textcolor{red}{\phi(x)}}{}\ . \tag{9}$$

Thus we have obtained a well-defined non-genuine local operator lying at the end of a line operator.

---

Most importantly, non-genuine operators form a layered structure:

1. We begin with genuine $q$-dimensional operators, which we denote by $\mathcal{O}_q^{(x)}$ with the superscript $x$ distinguishing different such operators.

2. Given an ordered pair $(\mathcal{O}_q^{(a)}, \mathcal{O}_q^{(b)})$ of two $q$-dimensional operators, we can have $(q-1)$-dimensional non-genuine operators changing $\mathcal{O}_q^{(a)}$ to $\mathcal{O}_q^{(b)}$, which we denote as $\mathcal{O}_{q-1}^{(a,b;x)}$ with the superscript $x$ distinguishing different such operators.

3. Given an ordered pair $(\mathcal{O}_{q-1}^{(a,b;A)}, \mathcal{O}_{q-1}^{(a,b;B)})$ of two $(q-1)$-dimensional operators changing $\mathcal{O}_q^{(a)}$ to $\mathcal{O}_q^{(b)}$, we can have $(q-2)$-dimensional non-genuine operators changing $\mathcal{O}_{q-1}^{(a,b;A)}$ to $\mathcal{O}_{q-1}^{(a,b;B)}$, which we denote as $\mathcal{O}_{q-2}^{(a,b;A,B;x)}$ with the superscript $x$ distinguishing different such operators.

4. We can continue in this fashion until we reach non-genuine local (i.e. 0-dimensional) operators.

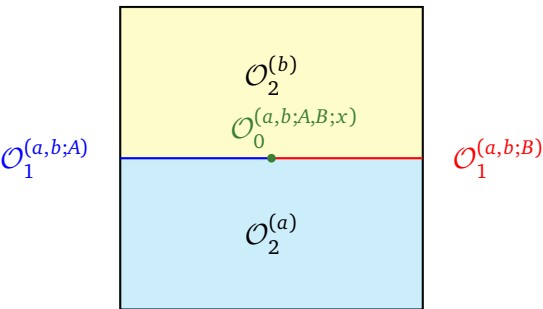

Figure 1: The layer structure of genuine (2d $\mathcal{O}_2$) and non-genuine ($\mathcal{O}_1$ and $\mathcal{O}_0$) operators as naturally occurring in theories with $d \geq 3$.

For $q = 2$, this layered structure of genuine and non-genuine operators can be depicted as in figure 1.

A similar layered structure is formed by morphisms and higher-morphisms in a higher-category:

---

**Definition: Higher-categories**

A $(q + 1)$-category comprises of the following data:

1. A set of objects or 0-morphisms, which we denote by $M_0^{(x)}$ with the superscript $x$ distinguishing different objects.

2. Given an ordered pair $(M_0^{(a)}, M_0^{(b)})$ of two objects, a set $\mathrm{Hom}(M_0^{(a)}, M_0^{(b)})$ of 1-morphisms from the object $M_0^{(a)}$ to the object $M_0^{(b)}$. We denote such 1-morphisms by $M_1^{(a,b;x)}$ with the superscript $x$ distinguishing different such 1-morphisms. For a usual category the story ends here, but for higher-categories it continues further as follows.

3. Given an ordered pair $(M_1^{(a,b;A)}, M_1^{(a,b;B)})$ of two 1-morphisms in $\mathrm{Hom}(M_0^{(a)}, M_0^{(b)})$, a set $\mathrm{Hom}(M_1^{(a,b;A)}, M_1^{(a,b;B)})$ of 2-morphisms from the 1-morphism $M_1^{(a,b)}$ to the 1-morphism $M_1^{(a,b)'}$. We denote such 2-morphisms by $M_2^{(a,b;A,B;x)}$ with the superscript $x$ distinguishing different such 2-morphisms.

4. Continuing iteratively in the above fashion, given an ordered pair $(M_{r-1}, M'_{r-1})$ of two $(r-1)$-morphisms in $\mathrm{Hom}(M_{r-2}, M'_{r-2})$, a set $\mathrm{Hom}(M_{r-1}, M'_{r-1})$ of $r$-morphisms from the $(r-1)$-morphism $M_{r-1}$ to the $(r-1)$-morphism $M'_{r-1}$. In this way, in a $(q+1)$-category we have $r$-morphisms for $0 \leq r \leq q+1$.

---

For $q = 1$, i.e. 2-categories, this layered structure can be depicted as in figure 2.

Given that layering structure of genuine and non-genuine operators is so similar to the layering structure inherent in the mathematics of higher-categories, one might then wonder whether genuine and non-genuine higher-charges can be combined into the structure of a higher-category. This indeed turns out to be the case. In order to motivate what this higher-category should be, let us first note that $(q + 1)$-representations, which as discussed above describe genuine $q$-charges, form objects of a $(q+1)$-category. We denote these $(q+1)$-categories

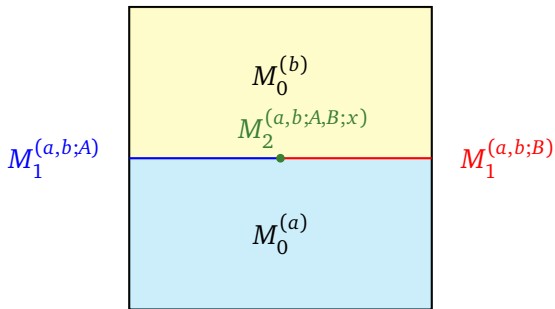

Figure 2: The layered structure for 2-category, which parallels the layer structure of operators in figure 1.

depending on the type of invertible symmetry as:

$$
\begin{aligned}
G^{(0)} \text{ 0-form symmetry}: && (q+1)\text{-Rep}(G^{(0)}), & \\
G^{(p)} \text{ } p\text{-form symmetry}: && (q+1)\text{-Rep}(\mathbb{G}^{(p+1)}_{G^{(p)}}), & \quad (10)\\
\mathbb{G}^{(p)} \text{ } p\text{-group symmetry}: && (q+1)\text{-Rep}(\mathbb{G}^{(p)}). &
\end{aligned}
$$

Indeed, for the simplest case $p = q = 0$, it is well known that representations of a group are objects of a category (also referred to as 1-category).

Thus, we are led to propose the following statement, whose various special sub-cases are justified in the bulk of this paper. The following statement is for a general $p$-group $\mathbb{G}^{(p)}$, but the reader can easily recover statements for a $G^{(0)}$ 0-form symmetry by simply substituting $p = 1$ and $\mathbb{G}^{(p=1)} = G^{(0)}$, and for a $G^{(p-1)}$ $(p-1)$-form symmetry by simply substituting $\mathbb{G}^{(p)} = \mathbb{G}^{(p)}_{G^{(p-1)}}$ to be the $p$-group associated to the $(p-1)$-form group $G^{(p-1)}$.

Table 1: Correspondence between the layering structure of higher-categories and layering structure of non-genuine higher-charges. Note that the layer formed by $(q+1)$-morphisms does not participate in this correspondence.

| Higher $(q+1)$-Category | Higher-Charges |
|---|---|
| Objects | Genuine $q$-dimensional operators |
| 1-Morphisms | Non-genuine $(q-1)$-dimensional operators |
| $r$-Morphisms; $r \leq q$ | Non-genuine $(q-r)$-dimensional operators |

---

**Statement 1.5: Non-genuine generalized charges**

$(q-r)$-charges of $(q-r)$-dimensional operators that can be embedded within genuine $q$-dimensional operators are $r$-morphisms in the $(q+1)$-category $(q+1)$-Rep$(\mathbb{G}^{(p)})$. In more detail, we have the following correspondence:

1. Consider two genuine $q$-dimensional operators $\mathcal{O}_q^{(a)}, \mathcal{O}_q^{(b)}$ with $q$-charges associated to objects $M_0^{(a)}, M_0^{(b)}$ of $(q+1)$-Rep$(\mathbb{G}^{(p)})$. The $(q-1)$-charges of $(q-1)$-dimensional operators changing $\mathcal{O}_q^{(a)}$ into $\mathcal{O}_q^{(b)}$ are elements of the set of 1-morphisms Hom$(M_0^{(a)}, M_0^{(b)}) \subset (q+1)$-Rep$(\mathbb{G}^{(p)})$.

2. Consider two $(q-1)$-dimensional operators $\mathcal{O}_{q-1}^{(a,b;A)}, \mathcal{O}_{q-1}^{(a,b;B)}$ with $(q-1)$-charges associated to 1-morphisms $M_1^{(a,b;A)}, M_1^{(a,b;B)} \in$ Hom$(M_0^{(a)}, M_0^{(b)}) \subset (q+1)$-Rep$(\mathbb{G}^{(p)})$. The $(q-2)$-charges of $(q-2)$-dimensional operators changing $\mathcal{O}_{q-1}^{(a,b;A)}$ into $\mathcal{O}_{q-1}^{(a,b;B)}$ are elements of the set of 2-morphisms Hom$(M_1^{(a,b;A)}, M_1^{(a,b;B)}) \subset (q+1)$-Rep$(\mathbb{G}^{(p)})$.

3. Continuing iteratively in the above fashion, consider two $(q-r+1)$-dimensional operators $\mathcal{O}_{q-r+1}, \mathcal{O}'_{q-r+1}$ with $(q-r+1)$-charges associated to $(r-1)$-morphisms $M_{r-1}, M'_{r-1} \in$ Hom$(M_{r-2}, M'_{r-2}) \subset (q+1)$-Rep$(\mathbb{G}^{(p)})$. The $(q-r)$-charges of $(q-r)$-dimensional operators changing $\mathcal{O}_{q-r+1}$ into $\mathcal{O}'_{q-r+1}$ are elements of the set of $r$-morphisms Hom$(M_{r-1}, M'_{r-1}) \subset (q+1)$-Rep$(\mathbb{G}^{(p)})$.

Note that the last level of morphisms, i.e. $(q+1)$-morphisms of $(q+1)$-Rep$(\mathbb{G}^{(p)})$, does not make an appearance at the level of generalized charges. Instead the last level of generalized charges, which is that of non-genuine 0-charges, is described by $q$-morphisms of $(q+1)$-Rep$(\mathbb{G}^{(p)})$. We summarize the correspondence between higher-charges and higher-morphisms in table 1.

---

## 1.4 Twisted generalized charges

Finally, let us note that the statements of the previous subsection are valid only if the non-genuine operators furnishing higher-charges are not in twisted sectors for the symmetry, which are defined as follows:

---

**Definition: Twisted Sectors**

We say that an operator lies in a twisted sector of the symmetry $\mathbb{G}^{(p)}$ if it lies at the end of one of the following types of topological operators related to the symmetry $\mathbb{G}^{(p)}$:

1. **Symmetry generators**: These are topological operators generating the $r$-form symmetry groups inside the $p$-group $\mathbb{G}^{(p)}$ for $0 \leq r \leq p-1$.

2. **Condensation defects**: These are topological operators obtained by gauging the above symmetry generators on positive codimensional sub-manifolds in spacetime [16, 62].

---

The reason for the inclusion of condensation defects is because we want to discuss operators living at the ends of topological operators in the symmetry fusion $(d-1)$-category

$$(d-1)\text{-Vec}(\mathbb{G}^{(p)}),\tag{11}$$

that arises whenever we have $\mathbb{G}^{(p)}$ symmetry in a $d$-dimensional QFT. See [17] for more details on higher-categories associated to symmetries, and [46] or the end of appendix B for more details on the higher-categories (11) associated to invertible symmetries. The elements of this $(d-1)$-category include both symmetry generators and condensation defects.

In this paper, we will study in detail twisted $q$-charges for a $G^{(0)}$ 0-form symmetry lying in twisted sectors associated to symmetry generators of $G^{(0)}$. Here we will encounter the following statement:

---

**Statement 1.6: Twisted generalized charges for 0-form symmetries**

Let $D_{d-1}^{(g)}$ be a codimension-1 topological operator generating $g \in G^{(0)}$. Then the higher-charges of codimension-$n$ operators, $n \geq 2$, that can be placed at the boundary of $D_{d-1}^{(g)}$ are valued in the $(d-1)$-category

$$(d-1)\text{-Rep}^{[\omega_g]}(H_g). \tag{12}$$

This category has

$$\text{Objects} = \left\{ [\omega_g] - \text{twisted } (d-1)\text{-representations of the centralizer } H_g \subseteq G^{(0)} \text{ of } g \right\}, \tag{13}$$

where

$$[\omega_g] \in H^d(H_g, \mathbb{C}^\times), \tag{14}$$

is obtained from the information of the 't Hooft anomaly $[\omega] \in H^{d+1}(G^{(0)}, \mathbb{C}^\times)$ for the $G^{(0)}$ 0-form symmetry and the symmetry element $g \in G^{(0)}$ by performing what is known as a slant product. See section 5.2 for more details.

---

Expanding out this statement in more detail, we have

1. $(d-2)$-charges of operators lying at the boundary of $D_{d-1}^{(g)}$ are $[\omega_g]$-twisted $(d-1)$-representations of $H_g$, or in other words 0-morphisms in $(d-1)\text{-Rep}^{[\omega_g]}(H_g)$.

2. For $1 \leq r \leq d-2$, we have that the $(d-r-2)$-charges going from a $(d-r-1)$-charge described by an $(r-1)$-morphism $M_{r-1} \in (d-1)\text{-Rep}^{[\omega_g]}(H_g)$ to a $(d-r-1)$-charge described by an $(r-1)$-morphism $M'_{r-1} \in (d-1)\text{-Rep}^{[\omega_g]}(H_g)$, are described by the $r$-morphisms from $M_{r-1}$ to $M'_{r-1}$ in $(d-1)\text{-Rep}^{[\omega_g]}(H_g)$.

The above statement only incorporates the action of the centralizer $H_g$. Consider a $(d-r-2)$-dimensional $g$-twisted sector operator $\mathcal{O}_{d-r-2}$ carrying a $(d-r-2)$-charge described by an $r$-morphism $M_r \in (d-1)\text{-Rep}^{[\omega_g]}(H_g)$. Acting on $\mathcal{O}_{d-r-2}$ by an element $h \notin H_g$, we obtain a $(d-r-2)$-dimensional operator $\mathcal{O}'_{d-r-2}$ in $hgh^{-1}$-twisted sector carrying an isomorphic $(d-r-2)$-charge described by an $r$-morphism $M'_r \in (d-1)\text{-Rep}^{[\omega_{hgh^{-1}}]}(H_{hgh^{-1}})$.

We will also study twisted generalized charges in a couple of other contexts:

- Twisted 1-charges in 4d when there is mixed 't Hooft between 1-form and 0-form symmetries. This reproduces a hallmark action of non-invertible 0-form symmetries on line operators in 4d theories.

- Generalized charges for operators lying in twisted sectors associated to condensation defects of non-anomalous higher-group symmetries. An interesting physical phenomena that arises in this context is the conversion of a non-genuine operator in a twisted sector associated to a condensation defect to a genuine operator and vice versa, which induces

maps between twisted and untwisted generalized charges. In the language of [63], this can be phrased as the relationship between *relative* and *absolute* defects in an *absolute* theory. In Part II of this series of papers [1], we will upgrade this relationship to incorporate *relative* defects in *relative* theories.

Part II [1] will also extend the discussion to non-invertible, or more generally, categorical symmetries. The central tool to achieve this is the Drinfeld center of a given higher fusion category. We will formulate the current paper in the context of the Drinfeld center and see how this allows a generalization to non-invertible symmetries and their action on charged objects.

## 1.5  Organization of the paper

This paper is organized as follows.

Section 2 discusses generalized charges for standard global symmetries, also known as 0-form symmetries. The aim of this section is to justify the statement 1.2. After reviewing in section 2.1 why 0-charges are described by representations of the 0-form symmetry group, we encounter the first non-trivial statement of this paper in section 2.2, where it is argued that 1-charges are described by 2-representations of the 0-form symmetry group. The arguments are further generalized in section 2.3 to show that $q$-charges are described by $(q + 1)$-representations of the 0-form symmetry group. An important physical phenomenon exhibited by $q$-charges for $q \geq 2$ is that of symmetry fractionalization, which we discuss in detail with various examples.

Section 3 discusses generalized charges for 1-form symmetries. The aim of this section is to justify the statement 1.3, at least for $p = 1$. In section 3.1, we review why 1-charges are described by 2-representations of the 2-group associated to the 1-form symmetry group, which coincide with representations of the 1-form symmetry group. In section 3.2, we discuss 2-charges under 1-form symmetries. These exhibit many interesting physical phenomena involving localized symmetries (which are possibly non-invertible), induced 1-form symmetries, and interactions between localized and induced symmetries. Ultimately, we argue that all these physical phenomena are neatly encapsulated as information describing a 3-representation of the 2-group associated to the 1-form symmetry group. In section 3.3, we briefly discuss $q$-charges of 1-form symmetries for $q \geq 3$, which in addition to the physical phenomena exhibited by 2-charges, also exhibit symmetry fractionalization.

Section 4 discusses non-genuine generalized charges. The aim of this section is to justify the statement 1.5. In section 4.1, we study non-genuine 0-charges of 0-form symmetries and argue that they are described by intertwiners between 2-representations of the 0-form symmetry group. In section 4.2, we discuss non-genuine 0-charges under 1-form symmetries, which recovers the well-known statement that two line operators with different charges under a 1-form symmetry cannot be related by screening. This is consistent with the corresponding mathematical statement that there are no intertwiners between two different irreducible 2-representations of the 2-group associated to a 1-form symmetry group.

Section 5 discusses twisted generalized charges. The aim of this section is to justify the statement 1.6 and it explores a few more situations. In section 5.1, we study generalized charges formed by operators living in twisted sectors correspond to symmetry generators of a non-anomalous 0-form symmetry. We argue that $g$-twisted generalized charges are described by higher-representations of the stabilizer $H_g$ of $g$. In section 5.2, we allow the 0-form symmetry to have anomaly. The $g$-twisted generalized charges are now described by twisted higher-representations of $H_g$. In section 5.3, we study 1-charges formed by line operators lying in twisted sector of a 1-form symmetry in 4d, where there is additionally a 0-form symmetry with a mixed 't Hooft anomaly with the 1-form symmetry. This situation arises in 4d $\mathcal{N} = 1$ and

$\mathcal{N} = 4$ pure Yang-Mills theories. We find that in such a situation the 0-form symmetry necessarily must permute the charge of the line operator under the 1-form symmetry, a fact that is tied to the action of non-invertible duality defects on lines. Finally, in section 5.4, we study operators lying in twisted sectors of condensation defects. These operators can be mapped to untwisted sector operators, and untwisted sector operators having certain induced symmetries can be converted into twisted sector operators for condensation defects. These maps relate untwisted and twisted generalized charges of these operators.

Section 6 discusses 1-charges for general 2-group symmetries, where we include both the action of 0-form on 1-form symmetries along with the possibility of a non-trivial Postnikov class. The aim of this section is to justify the statement 1.4, at least for $p = 2, q = 1$.

The conclusions, along with an outlook, are presented in section 7. Lastly, we have a couple of appendices. We collect some important notation and terminology in appendix A. In appendix B, we discuss the mathematical definition of higher-representations of groups and higher-groups.

Before we begin the main text of the paper, let us make a technical disclaimer aimed mostly at experts.

---

**Disclaimer: Restrictions on dimensions of operators and 't Hooft anomalies**

Throughout this paper,

1. We consider action of $\mathbb{G}^{(p)}$ only on operators of co-dimension at least 2.

2. We allow $\mathbb{G}^{(p)}$ to have a 't Hooft anomaly associated to an element of $H^{d+1}(B\mathbb{G}^{(p)}, \mathbb{C}^\times)$, where $B\mathbb{G}^{(p)}$ denotes the classifying space of $\mathbb{G}^{(p)}$.

Both these assumptions go hand in hand. The above type of 't Hooft anomaly is localized only on points, and so does not affect the action of $\mathbb{G}^{(p)}$ on untwisted sector operators of co-dimension at least 2. Thus, while discussing the action of $\mathbb{G}^{(p)}$ on operators of these co-dimensions, we can effectively forget about the anomaly. It should be noted though that the anomaly does affect the action of $\mathbb{G}^{(p)}$ on twisted sector operators of co-dimension 2. We discuss in detail the modification caused by an anomaly for a 0-form symmetry.

---

## 2 Generalized charges for 0-form symmetries

In this section we study physically the action of a 0-form symmetry group $G^{(0)}$ on operators of various dimensions, and argue that $q$-charges of these operators are $(q+1)$-representations of $G^{(0)}$, justifying the statement 1.2.

### 2.1 0-charges

Here we reproduce the well-known argument for the following piece of statement 1.2.

---

**Statement 2.1: 0-charges for 0-form symmetries**

0-charges of a $G^{(0)}$ 0-form symmetry are representations of the group $G^{(0)}$.

---

The action on local operators is obtained by moving the codimension-1 topological operators generating the 0-form symmetry across the location where local operators are inserted.

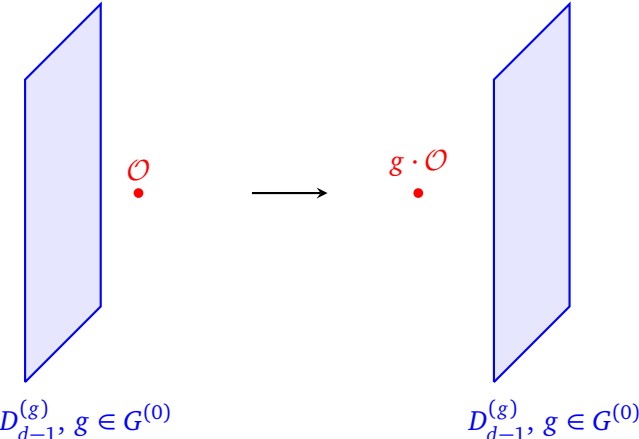

Figure 3: Action of a 0-form symmetry realized by a codimension-1 topological defect $D_{d-1}^{(g)}$, $g \in G^{(0)}$, on a local operator $\mathcal{O}$.

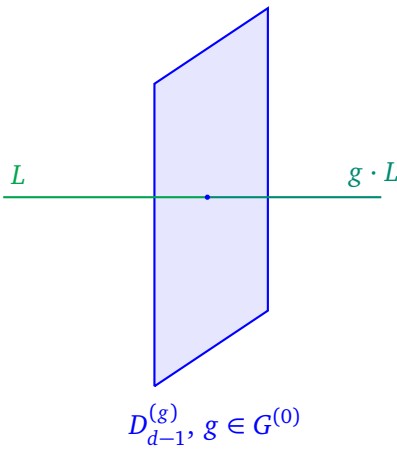

Figure 4: A 0-form symmetry $g \in G^{(0)}$ may act by changing a line operator $L$ to another line operator $g \cdot L$.

See figure 3. Moving a topological operator labeled by $g \in G^{(0)}$ across, transforms a local operator $\mathcal{O}$ as

$$g : \qquad \mathcal{O} \longrightarrow g \cdot \mathcal{O} \,. \tag{15}$$

One can now move two topological operators across sequentially, or first fuse them and then move them across, leading to the consistency condition

$$g_2 \cdot (g_1 \cdot \mathcal{O}) = (g_2 g_1) \cdot \mathcal{O} \,. \tag{16}$$

Moreover, the topological operator corresponding to $1 \in G^{(0)}$ is the identity operator, which clearly has the action

$$1 : \qquad \mathcal{O} \longrightarrow \mathcal{O} \,. \tag{17}$$

Consequently, 0-charges furnished by local operators are representations of $G^{(0)}$.

## 2.2 1-charges

In this subsection we study 1-charges of 0-form symmetries, i.e. the possible actions of 0-form symmetries on line operators. Similar analyses have appeared recently in [64, 65].

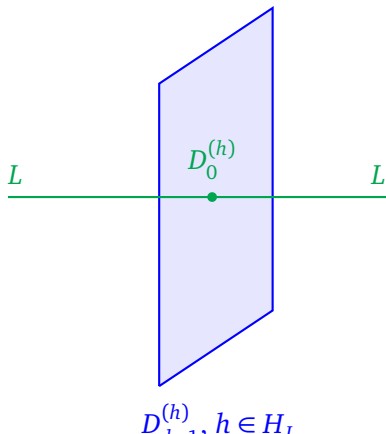

Figure 5: A topological local operator $D_0^{(h)}$ arising at the junction of a line operator $L$ and the 0-form symmetry generator $D_{d-1}^{(h)}, h \in H_L$. Such topological local operators $D_0^{(h)}$ for all $h \in H_L$ generate an induced 0-form symmetry on $L$.

Consider now a simple line operator[1] $L$, where simplicity of an operator is defined as follows.

---

**Definition**

A $q$-dimensional operator $\mathcal{O}_q$ is called **simple** if the vector space formed by topological operators living on the world-volume of $\mathcal{O}_q$ is one-dimensional. That is, the only non-zero topological local operators living on $\mathcal{O}_q$ are $\mathbb{C}^\times$ multiples of the identity local operator on $\mathcal{O}_q$.

---

As $L$ passes through a codimension-1 topological operator labeled by $g \in G^{(0)}$, it may get transformed into a different line operator $g \cdot L$. See figure 4. This means that $L$ lives in a multiplet (or orbit)

$$M_L = \left\{ g \cdot L; \ g \in G^{(0)} \right\}, \tag{18}$$

of line operators that are related to each other by the action of $G^{(0)}$.

Let $H_L \subseteq G^{(0)}$ be the subgroup that leaves the line operator $L$ invariant (i.e. the stabilizer group)

$$H_L = \left\{ h \in G^{(0)}; \ h \cdot L = L \right\}. \tag{19}$$

Then we can label different line operators in the multiplet $M_L$ by right cosets of $H_L$ in $G^{(0)}$, i.e.

$$M_L = \left\{ [g] = H_L g; \ g \in G^{(0)} \right\}. \tag{20}$$

Consequently, we denote a line operator in $M_L$ obtained by acting $g \in G^{(0)}$ on $L$ by $L_{[g]}$, and the action of $g$ is

$$g: \qquad L \longrightarrow g \cdot L = L_{[g]}, \qquad [g] \equiv H_L g. \tag{21}$$

Let us now look more closely at the action of $H_L$ on $L$, which leaves the line operator invariant. $H_L$ can be understood as an induced 0-form symmetry on the line operator $L$, which is a 0-form symmetry generated by the topological local operators $D_0^{(h)}$ arising at the junction of $L$ with codimension-1 topological operators associated to $H_L$ extending into the bulk $d$-dimensional spacetime. See figure 5.

---

[1] In this paper, the term '$q$-dimensional operator' almost always refers to a simple $q$-dimensional operator.

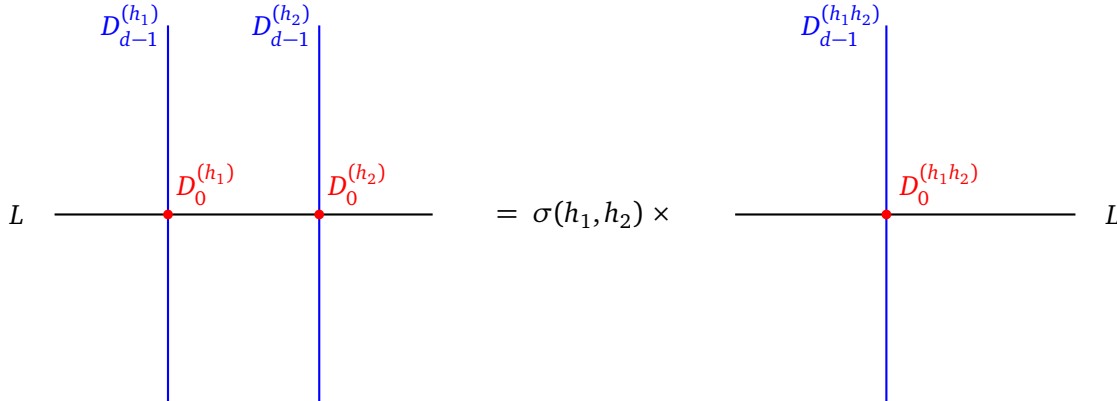

Figure 6: Fusion of induced 0-form symmetry generators $D_0^{(h_i)}$ on a line operator $L$.

This induced $H_L$ 0-form symmetry may have a 't Hooft anomaly, which is described by an element of the group cohomology

$$[\sigma] \in H^2(H_L, \mathbb{C}^\times). \tag{22}$$

This means that the fusion of the induced 0-form symmetry generators $D_0^{(h_1)}$ and $D_0^{(h_2)}$ differs from $D_0^{(h_1 h_2)}$ as follows

$$D_0^{(h_1)} D_0^{(h_2)} = \sigma(h_1, h_2) D_0^{(h_1 h_2)}, \qquad h_i \in H_L, \tag{23}$$

where $\sigma$ is a representative of the class $[\sigma]$, and so $\sigma(h_1, h_2)$ is a $\mathbb{C}^\times$ factor in the above equation. See figure 6. We do have the freedom of rescaling[2] $D_0^{(h)}$ independently for all $h \in H_L$, but this only modifies the $\mathbb{C}^\times$ factors in (23) by replacing $\sigma$ by a different representative $\sigma'$ of $[\sigma]$.

We can also study similar properties for some other line operator $L_{[g]}$ in the multiplet $M_L$. It is easy to see that the induced 0-form symmetry on $L_{[g]}$ is $H_{L_{[g]}} := g H_L g^{-1}$ which is isomorphic to $H_L$ and the 't Hooft anomaly of the $H_{L_{[g]}}$ induced 0-form symmetry is obtained by simply transporting $[\sigma]$ by the isomorphism between $H_L$ and $H_{L_{[g]}}$. Thus, there is no new information contained in the action of $G^{(0)}$ on $L_{[g]}$ beyond the pair $(H_L, [\sigma])$.

The pair $(H_L, [\sigma])$ *precisely* defines an irreducible 2-representation of $G^{(0)}$ [26, 27, 66]! See appendix B for mathematical definition of 2-representations. Hence, we have justified the following piece of statement 1.2.

---

**Statement 2.2: 1-charges for 0-form symmetries**

1-charges of a $G^{(0)}$ 0-form symmetry are 2-representations of the group $G^{(0)}$.

---

In more detail, there are three fundamental properties of a 1-charge associated to a 2-representation (which we denote with an appropriate superscript)

$$\rho^{(2)} = (H_L, [\sigma]), \tag{24}$$

which capture the action of $G^{(0)}$ on line operators furnishing the 1-charge $\rho^{(2)}$:

---

[2] Said differently, the junction of $L$ with $D_{d-1}^{(h)}$ contains a one-dimensional vector space of junction operators, and the $D_0^{(h)}$ junction operators used above are some random choice of vectors in these vector spaces. The rescalings correspond to making another choice for junction operators $D_0^{(h)}$.

1. **Size of the multiplet**: Line operators furnishing the 1-charge $\rho^{(2)}$ lie in a multiplet $M_L$, with total number of lines in $M_L$ being

$$n := \text{Size of the multiplet } M_L = \text{Number of right cosets of } H_L \text{ in } G^{(0)}. \quad (25)$$

   The multiplet is irreducible in the sense that we can obtain any line $L' \in M_L$ from any other line $L \in M_L$ by acting by some element $g \in G^{(0)}$.
   The size $n$ of $M_L$ is known as the **dimension** of the 2-representation $\rho^{(2)}$.

2. **Induced 0-form symmetry**: The subgroup $H_L \subseteq G^{(0)}$ forming part of the data of the 1-charge $\rho^{(2)}$ specifies the global 0-form symmetry induced on a particular line operator $L$ in the multiplet $M_L$.

3. **'t Hooft anomaly of induced symmetry**: The element $[\sigma] \in H^2(H_L, \mathbb{C}^\times)$ forming part of the data of the 1-charge $\rho^{(2)}$ specifies the 't Hooft anomaly of the $H_L$ induced 0-form symmetry of $L$.

We can also express the fact that a multiplet $M_L$ of line operators furnishes a 1-charge $\rho^{(2)}$ of $G^{(0)}$ 0-form symmetry by saying that the *non-simple* line operator

$$\bigoplus_{L' \in M_L} L', \quad (26)$$

transforms in the 2-representation $\rho^{(2)}$ of $G^{(0)}$.

   Let us now consider concrete field theory examples of line operators forming 2-representations of a 0-form symmetry.

---

**Example 2.1: Simplest example of non-trivial 1-charge**

The simplest non-trivial irreducible 2-representation arises for

$$G^{(0)} = \mathbb{Z}_2, \quad (27)$$

which is

$$\rho^{(2)} = (H = 1, [\sigma] = 0). \quad (28)$$

To realize it physically, we need a multiplet of two simple line operators $L$ and $L'$, or in other words a non-simple line operator

$$L \oplus L'. \quad (29)$$

The action of (27) then exchanges $L$ and $L'$. In fact, $\mathbb{Z}_2$ has only two irreducible 2-representations, with the other 2-representation being the trivial one

$$\rho^{(2)} = (H = \mathbb{Z}_2, [\sigma] = 0), \quad (30)$$

which is physically realized on a single simple line operator

$$L, \quad (31)$$

which is left invariant by the action of (27).

   Both of these 2-representations arise in the following example Quantum Field Theory. Take

$$\mathfrak{T} = d\text{-dimensional pure Spin}(2N) \text{ gauge theory.} \quad (32)$$

This theory has (27) 0-form symmetry arising from the outer-automorphism of the gauge algebra $\mathfrak{so}(2N)$. The non-trivial 2-representation is furnished by

$$W_S \oplus W_C \,, \tag{33}$$

where $W_S$ is Wilson line in irreducible spinor representation $S$ of $\mathfrak{so}(2N)$ and $W_C$ is Wilson line in irreducible cospinor representation $C$ of $\mathfrak{so}(2N)$. Indeed, the outer-automorphism exchanges the spinor and cospinor representations, and hence the two Wilson lines are exchanged

$$W_S \longleftrightarrow W_C \,, \tag{34}$$

under the action of (27).

On the other hand, there are representations of $\mathfrak{so}(2N)$ left invariant by the outer-automorphism, e.g. the vector representation $V$. The corresponding Wilson lines are left invariant by (27), e.g. we have

$$W_V \phantom{xxxxxx} \tag{35}$$

and hence $W_V$ transforms in the trivial 2-representation of (27).

In the above example, neither of the 1-charges involved a 't Hooft anomaly for the 0-form symmetry induced on the line operators furnishing the 1-charge. Below we discuss an example where there is a non-trivial 't Hooft anomaly.

**Example 2.2: 1-charges having anomalous induced symmetry**

The simplest group $G^{(0)}$ having non-trivial $H^2(G^{(0)}, \mathbb{C}^\times)$ is

$$G^{(0)} = \mathbb{Z}_2 \times \mathbb{Z}_2 \,, \tag{36}$$

for which we have

$$H^2(\mathbb{Z}_2 \times \mathbb{Z}_2, \mathbb{C}^\times) = \mathbb{Z}_2 \,. \tag{37}$$

Here we present an example of a QFT $\mathfrak{T}$ which contains a line operator $L$ on which all of the bulk (36) 0-form symmetry descends to an induced 0-form symmetry $H_L = \mathbb{Z}_2 \times \mathbb{Z}_2$ along with the 't Hooft anomaly $[\sigma]$ for the induced 0-form symmetry being given by the non-trivial element of (37). In other words, $L$ transforms in the 2-representation

$$\rho^{(2)} = (\mathbb{Z}_2 \times \mathbb{Z}_2, [\sigma] \neq 0) \,, \tag{38}$$

of (36).

For this purpose, we take

$$\mathfrak{T} = \text{3d pure gauge theory with gauge group } SO(4N) \,. \tag{39}$$

This theory has a 1-form symmetry

$$G^{(1)} = \mathbb{Z}_2^{(1)} \,, \tag{40}$$

which arises from the center of the gauge group $SO(4N)$. The theory also has a 0-form symmetry

$$G^{(0)} = \mathbb{Z}_2^m \times \mathbb{Z}_2^o. \tag{41}$$

The $\mathbb{Z}_2^m$ 0-form symmetry, also known as the magnetic symmetry, acts non-trivially on monopole operators inducing monopole configurations (on a small sphere $S^2$ around the operator) of $SO(4N)$ that cannot be lifted to monopole configurations of Spin$(4N)$. On the other hand, the $\mathbb{Z}_2^o$ 0-form symmetry arises from the outer-automorphism of the $\mathfrak{so}(4N)$ gauge algebra.

The 1-charge associated to the 2-representation (38) is furnished by any solitonic line defect for the 1-form symmetry [67,68], i.e. any line defect which induces a non-trivial background for 1-form symmetry on a small disk intersecting the line at a point:

$$L \quad \underline{\hspace{2cm}} \bullet \underline{\hspace{2cm}} . \tag{42}$$

This is a consequence of **anomaly inflow**: in the bulk we have the following mixed anomaly between $G^{(0)}$ and $G^{(1)}$

$$\mathcal{A}_4 = \exp\left( i\pi \int B_2 \cup A_1^{(m)} \cup A_1^{(o)} \right), \tag{43}$$

where $B_2$ is the background field for $\mathbb{Z}_2^{(1)}$ 1-form symmetry and $A_1^{(i)}$ are background fields for the $\mathbb{Z}_2^i$ 0-form symmetries. This anomaly flows to an anomaly

$$\mathcal{A}_2 = \exp\left( i\pi \int A_{1,L}^{(m)} \cup A_{1,L}^{(o)} \right), \tag{44}$$

on such a solitonic line defect $L$, where $A_{1,L}^{(i)}$ are restrictions along $L$ of the background fields $A_1^{(i)}$.

The simplest example of such a solitonic line defect is

$$L = D_1^{(-)} := \text{Topological line operator generating } \mathbb{Z}_2^{(1)} \text{ 1-form symmetry.} \tag{45}$$

In this case an independent check that this topological line operator $D_1^{(-)}$ carries an induced anomaly was performed in [47], by showing that $\mathbb{Z}_2^m$ and $\mathbb{Z}_2^o$ actions anti-commute along $D_1^{(-)}$, which implies the 't Hooft anomaly (44).

Examples of non-topological solitonic line defects are obtained as fusion products

$$D_1^{(-)} \otimes W, \tag{46}$$

where $W$ can be taken to be any Wilson line operator. A line $D_1^{(-)} \otimes W$ is non-topological because a Wilson line $W$ is non-topological.

## 2.3 Higher-charges

We now consider the extension to higher-dimensional operators, i.e. $q$-charges for $q \geq 2$, for 0-form symmetries. There is a natural extension of the discussion in the last section on 1-charges, to higher-dimensions, which will be referred to as **higher-charges of group cohomology type**. However, we will see that higher-representation theory forces us to consider also a

generalization thereof, namely **non-group cohomology type**, which in fact have a natural physical interpretation as **symmetry fractionalization**.[3]

### 2.3.1 Higher-charges: Group cohomology type

There is an extremely natural generalization of the actions of $G^{(0)}$ on line operators to actions of $G^{(0)}$ on higher-dimensional operators.

These give rise to a special type of $q$-charges that we refer to as of **group cohomology type**, which are described by special types of $(q + 1)$-representations of the form

$$\rho^{(q+1)} = \big(H(\mathcal{O}_q), [\sigma]\big), \tag{47}$$

which is comprised of a subgroup

$$H(\mathcal{O}_q) \subseteq G^{(0)}, \tag{48}$$

and a cocycle

$$[\sigma] \in H^{q+1}\big(H(\mathcal{O}_q), \mathbb{C}^\times\big). \tag{49}$$

The action of $G^{(0)}$ is captured in this data quite similar to as in previous subsection.

1. **Size of the multiplet**: $q$-dimensional operators furnishing the $q$-charge $\rho^{(q+1)}$ lie in a multiplet $M(\mathcal{O}_q)$, with total number of $q$-dimensional operators in $M(\mathcal{O}_q)$ being

    $$n := \text{Size of the multiplet } M(\mathcal{O}_q) = \text{Number of right cosets of } H(\mathcal{O}_q) \text{ in } G^{(0)}. \tag{50}$$

    The multiplet is irreducible in the sense that we can obtain any $q$-dimensional operator $\mathcal{O}'_q \in M(\mathcal{O}_q)$ from any other $q$-dimensional operator $\mathcal{O}_q \in M(\mathcal{O}_q)$ by acting by some element $g \in G^{(0)}$.

    Mathematically, the size $n$ of $M(\mathcal{O}_q)$ is known as the **dimension** of the $(q + 1)$-representation $\rho^{(q+1)}$.

2. **Induced 0-form symmetry**: The subgroup $H(\mathcal{O}_q) \subseteq G^{(0)}$ forming part of the data of the $q$-charge $\rho^{(q+1)}$ specifies the global 0-form symmetry induced on a particular $q$-dimensional operator $\mathcal{O}_q$ in the multiplet $M(\mathcal{O}_q)$.

3. **'t Hooft anomaly of the induced symmetry**: The element $[\sigma] \in H^{q+1}\big(H(\mathcal{O}_q), \mathbb{C}^\times\big)$ forming part of the data of the $q$-charge $\rho^{(q+1)}$ specifies the 't Hooft anomaly of the $H(\mathcal{O}_q)$ induced 0-form symmetry of $\mathcal{O}_q$.

As we discussed in the previous subsection, for $q = 1$, the group cohomology type 1-charges are the most general 1-charges. However, for $q \geq 2$, the group cohomology type $q$-charges only form a small subset of all possible $q$-charges. We describe the most general 2-charges in the next subsection.

> **Example 2.3: Simplest example of a non-trivial $q$-charge**
>
> The simplest $(q + 1)$-representation of group cohomology type is
>
> $$\rho^{(q+1)} = (H = 1, [\sigma] = 0), \tag{51}$$

---

[3]We caution the reader that the symmetry fractionalization discussed here is different from the notion of symmetry fractionalization used in earlier works in the literature. The symmetry fractionalization appearing in this work is a *genuine* fractionalization of a symmetry to a bigger symmetry. On the other hand, the symmetry fractionalization appearing in other works relates to choices of possible couplings of the theory to backgrounds of a symmetry of the theory. For example, a theory with $G$ 0-form symmetry and $A$ 1-form symmetry has $H^2(G, A)$ worth of possible couplings to $G$ background fields, which are traditionally referred to as symmetry fractionalizations. We thank an anonymous referee for suggesting to stress this important distinction to us.

for $G^{(0)} = \mathbb{Z}_2$. This is physically realized by an exchange of two simple $q$-dimensional operators $\mathcal{O}_q$ and $\mathcal{O}'_q$.

This $q$-charge arises for the outer-automorphism 0-form symmetry in

$$\mathfrak{T} = (q + 2)\text{-dimensional pure Spin}(4N) \text{ gauge theory,} \tag{52}$$

and is carried by the topological codimension-2 operators $D_q^{(S)}$ and $D_q^{(C)}$, generating $\mathbb{Z}_2$ 1-form symmetries that do not act on spinor and cospinor representations respectively.

---

**Example 2.4: $q$-charges with anomalous induced symmetry: Anomaly inflow**

Examples of group cohomology type $q$-charges carrying non-trivial $[\sigma]$ can be obtained via anomaly inflow from the bulk $d$-dimensional QFT, as in an example discussed in the previous subsection. For example, the $q$-charge

$$\rho^{(q+1)} = (G^{(0)}, [\sigma]), \tag{53}$$

can be furnished by a $q$-dimensional solitonic defect inducing a background (on a small $(d-q)$-dimensional disk intersection its locus) for a $(d-q-1)$-form symmetry having a mixed 't Hooft anomaly with 0-form symmetry in the bulk roughly[a] of the form

$$B_{d-q} \cup \sigma, \tag{54}$$

where $B_{d-q}$ is the background field for the $(d-q-1)$-form symmetry.

---

[a]For brevity, we are suppressing many details that need to be specified for the following expression for the anomaly to make sense.

---

### 2.3.2 Higher-charges: (Non-invertible) symmetry fractionalization type

At first sight, one might think that group cohomology type $q$-charges provide all possible $q$-charges. There are at least two reasons for believing so:

1. First of all, the mathematical structure of group cohomology type $q$-charges is a nice, uniform generalization of the mathematical structure of general 1-charges.

2. Secondly, the mathematical data of group cohomology type $q$-charges described in the previous subsection seems to incorporate all of the relevant physical information associated to the action of $G^{(0)}$ 0-form symmetry on $q$-dimensional operators.

However, if one believes statement 1.2, then one should check whether $(q+1)$-representations are all of group cohomology type. It turns out that this is not the case for $q \geq 2$, for which generic $q$-charges are in fact not of this type. In this way, the mathematics of higher-representations forces us to seek new physical phenomena that only start becoming visible when considering the action of $G^{(0)}$ 0-form symmetry on a $q \geq 2$-dimensional operator $\mathcal{O}_q$.

In turn, physically we will see that these non-group cohomology type higher representations have concrete realizations in terms of symmetry fractionalization. Perhaps the most intriguing implication is that invertible symmetries can fractionalize into non-invertible symmetries, as we will see in the example of a $\mathbb{Z}_2^{(0)}$ fractionalizing into the Ising category.

**Localized and induced symmetries.** This new physical phenomenon is the existence of localized symmetries, namely symmetries of the operator $\mathcal{O}_q$ generated by topological operators

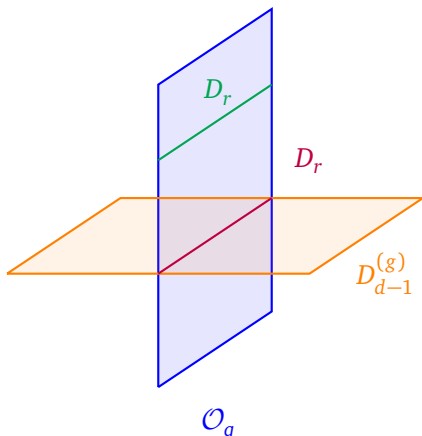

Figure 7: Localized (green) and induced localized (purple) symmetries on the operator $\mathcal{O}_q$. The latter arise in the world-volume occupied by the intersection of a bulk topological defect of codimension 1, $D_{d-1}^{(g)}$, and the operator $\mathcal{O}_q$.

living inside the worldvolume of $\mathcal{O}_q$. See figure 7. For $q = 1$, such localized symmetries correspond to the existence of topological local operators living on the line $\mathcal{O}_{q=1}$, but since $\mathcal{O}_{q=1}$ is taken to be simple, such localized symmetries are ruled out. For $q \geq 2$, even if $\mathcal{O}_q$ is taken to be simple, we can have topological operators of dimension $1 \leq r \leq q-1$ living inside the worldvolume of $\mathcal{O}_q$.

Additionally, we also have *induced* localized symmetries. These are generated by $(q-1)$-dimensional and lower-dimensional topological defects arising in the $(q-1)$-dimensional world-volume of a junction between $\mathcal{O}_q$ and a bulk codimension-1 topological defect $D_{d-1}^{(g)}$ generating $g \in G^{(0)}$. See figure 7.

> **Definition**
>
> We refer to localized symmetries induced by a bulk 0-form symmetry $g \in G^{(0)}$ as **induced localized symmetries in the $g$-sector**.

Then, induced localized symmetries in the identity sector are just the localized symmetries discussed in the previous paragraph.

Note that we can compose an induced localized symmetry in the $g$-sector with an induced localized symmetry in the $g'$-sector to obtain an induced localized symmetry in the $gg'$-sector.

**Mathematical structure.** As discussed in the introduction, the mathematical structure to encapsulate defects of various dimensions layered and embedded in each other is that of higher-categories. Thus, we can describe the induced localized symmetries in the $g$-sector by a (non-fusion) $(q-1)$-category

$$\mathcal{C}_g^{(q-1)}(\mathcal{O}_q). \tag{55}$$

In total, all induced localized symmetries are described by a $(q-1)$-category

$$\mathcal{C}^{(q-1)}(\mathcal{O}_q) := \bigoplus_{g \in G^{(0)}} \mathcal{C}_g^{(q-1)}(\mathcal{O}_q). \tag{56}$$

The composition of induced localized symmetries lying in different sectors discussed in the previous paragraph becomes a fusion structure on the $(q-1)$-category $\mathcal{C}^{(q-1)}(\mathcal{O}_q)$, converting

it into a fusion $(q-1)$-category. Moreover, since the fusion respects the group multiplication of the underlying sectors $\mathcal{C}^{(q-1)}(\mathcal{O}_q)$ is in fact a $G^{(0)}$**-graded fusion $(q-1)$-category**.

Now, $G^{(0)}$-graded fusion $(q-1)$-categories describe only 1-dimensional $(q+1)$-representations. This is because we have only been studying a special class of $q$-charges: a $q$-charge in this class describes a multiplet of size 1 of $q$-dimensional operators, i.e. there is only a single $q$-dimensional operator $\mathcal{O}_q$ in the multiplet. Indeed, while discussing above the structure of induced localized symmetries, we assumed that all the elements $g \in G^{(0)}$ leave $\mathcal{O}_q$ invariant. Allowing more $q$-dimensional operators to participate in the multiplet, we obtain $q$-charges described by more general $(q+1)$-representations that are described by $G^{(0)}$**-graded multi-fusion $(q-1)$-categories**. Thus, we have justified the statement 1.2 for general $q$.

**Symmetry fractionalization to a bigger 0-form group.** Non-group cohomology $q$-charges are associated to the physical phenomena of symmetry fractionalization, or more precisely the fractionalization of the bulk $G^{(0)}$ 0-form symmetry when induced on the operator $\mathcal{O}_q$. In general, the fractionalized induced symmetries are non-invertible, but for special classes of $q$-charges the fractionalized induced symmetry is again invertible. The invertible fractionalized induced symmetry can in general be a $q$-group symmetry. For the moment, let us focus on $q$-charges for which this $q$-group symmetry is just a 0-form symmetry $\widetilde{G}^{(0)}$.

Such a $q$-charge is specified by two pieces of data

1. A surjective homomorphism
$$\pi : \widetilde{G}^{(0)} \to G^{(0)} . \tag{57}$$

2. A choice of element
$$[\omega] \in H^{q+1}(\widetilde{G}^{(0)}, \mathbb{C}^{\times}) . \tag{58}$$

and is realized by a multiplet comprising of a single $q$-dimensional operator $\mathcal{O}_q$. The physical information of the $q$-charge is obtained from these two pieces of data as follows

1. **Localized symmetries**: There is a 0-form symmetry localized on the world-volume of $\mathcal{O}_q$ given by the kernel of $\pi$, $\ker(\pi) \subseteq \widetilde{G}^{(0)}$.

2. **Induced localized symmetries**: Additionally, we have induced localized symmetries. In the $g$-sector, these are in one-to-one correspondence with the elements of the subset
$$\pi^{-1}(g) \subseteq \widetilde{G}^{(0)} . \tag{59}$$

3. **Composition of induced localized symmetries**: The composition of induced localized symmetries is described by the group multiplication of $\widetilde{G}^{(0)}$.

   In other words, the bulk $G^{(0)}$ 0-form symmetry has **fractionalized** to a $\widetilde{G}^{(0)}$ induced 0-form symmetry on $\mathcal{O}_q$.

4. **'t Hooft anomaly of induced localized symmetries**: Finally, the element $[\omega]$ describes the 't Hooft anomaly of the $\widetilde{G}^{(0)}$ 0-form symmetry of $\mathcal{O}_q$.

Mathematically, these $q$-charges correspond to 1-dimensional $(q+1)$-representations whose associated $G^{(0)}$-graded fusion $(q-1)$-category $\mathcal{C}^{(q-1)}(\mathcal{O}_q)$ is

$$\mathcal{C}^{(q-1)}(\mathcal{O}_q) = (q-1)\text{-Vec}_{\widetilde{G}^{(0)}}^{[\omega]} , \tag{60}$$

of $\widetilde{G}^{(0)}$-graded $(q-1)$-vector spaces with a non-trivial coherence relation (also known as associator) described by the class $[\omega]$.

**Example 2.5: $G^{(0)} = \mathbb{Z}_2$ to $\widetilde{G}^{(0)} = \mathbb{Z}_4$ symmetry fractionalization**

A simple example of such a $q$-charge is provided by

$$G^{(0)} = \mathbb{Z}_2, \qquad \widetilde{G}^{(0)} = \mathbb{Z}_4, \tag{61}$$

where there is a unique possible surjective map $\pi$: it maps the two generators of $\mathbb{Z}_4$ to the generator of $\mathbb{Z}_2$. A $q$-dimensional operator $\mathcal{O}_q$ realizing this $q$-charge has a localized symmetry

$$\ker(\pi) = \mathbb{Z}_2. \tag{62}$$

That is, there is a $(q-1)$-dimensional topological operator $D^{(-)}_{q-1}$ living in the worldvolume of $\mathcal{O}_q$. Now let us look at induced localized symmetries lying in the non-trivial sector in $G^{(0)} = \mathbb{Z}_2$. These are in one-to-one correspondence with the two generators of $\widetilde{G}^{(0)} = \mathbb{Z}_4$. That is, there are two $(q-1)$-dimensional topological operators that can arise at the junction of $\mathcal{O}_q$ with the bulk codimension-1 topological defect $D_{d-1}$ that generates $G^{(0)} = \mathbb{Z}_2$. Let us denote these $(q-1)$-dimensional topological operators by

$$D^{(i)}_{q-1}, \qquad D^{(-i)}_{q-1}. \tag{63}$$

The statement of symmetry fractionalization is now as follows. We try to induce the $G^{(0)} = \mathbb{Z}_2$ symmetry on $\mathcal{O}_q$. In order to implement this symmetry on $\mathcal{O}_q$, we need to make a choice of a topological defect lying at the intersection of $\mathcal{O}_q$ and the symmetry generator $D_{d-1}$. We can either choose this topological defect to be $D^{(i)}_{q-1}$ or $D^{(-i)}_{q-1}$. Let us make the choice $D^{(i)}_{q-1}$ without loss of generality. Now we check whether the symmetry is still $\mathbb{Z}_2$ valued by performing the fusion of these topological defects. As we fuse $D_{d-1}$ with itself, it becomes a trivial defect, which means the symmetry is $\mathbb{Z}_2$-valued in the bulk. However along the worldvolume of $\mathcal{O}_q$ we have to fuse $D^{(i)}_{q-1}$ with itself, resulting in the generator of the $\mathbb{Z}_2$ symmetry localized along $\mathcal{O}_q$

$$D^{(i)}_{q-1} \otimes D^{(i)}_{q-1} = D^{(-)}_{q-1}. \tag{64}$$

See figure 8. Thus we see explicitly that the bulk $G^{(0)} = \mathbb{Z}_2$ symmetry fractionalizes to $\widetilde{G}^{(0)} = \mathbb{Z}_4$ symmetry when we try to induce it on $\mathcal{O}_q$. The same result is obtained if we instead try to induce the $G^{(0)} = \mathbb{Z}_2$ symmetry on $\mathcal{O}_q$ using $D^{(-i)}_{q-1}$.

An example of such a symmetry fractionalization is obtained in any QFT $\mathfrak{T}$ having a $\mathbb{Z}_2^{(d-2)}$ $(d-2)$-form symmetry and a $G^{(0)} = \mathbb{Z}_2^{(0)}$ 0-form symmetry, along with a mixed 't Hooft anomaly of the form

$$\mathcal{A}_{d+1} = \exp\left( i\pi \int B_{d-1} \cup A_1 \cup A_1 \right), \tag{65}$$

and no pure 't Hooft anomaly for $\mathbb{Z}_2^{(d-2)}$. QFTs having such a symmetry and anomaly structure are ubiquitous: simply take a $d$-dimensional QFT $\mathfrak{T}'$ which has a non-anomalous $\mathbb{Z}_4$ 0-form symmetry, and gauge the $\mathbb{Z}_2$ subgroup of this 0-form symmetry. The resulting QFT after gauging can be identified with $\mathfrak{T}$. The $\mathbb{Z}_2^{(d-2)}$ $(d-2)$-form symmetry of $\mathfrak{T}$ is obtained as the dual of the $\mathbb{Z}_2$ 0-form symmetry of $\mathfrak{T}'$ being gauged, and $G^{(0)} = \mathbb{Z}_2^{(0)}$ 0-form symmetry of $\mathfrak{T}$ is the residual $\mathbb{Z}_4/\mathbb{Z}_2$ 0-form symmetry.

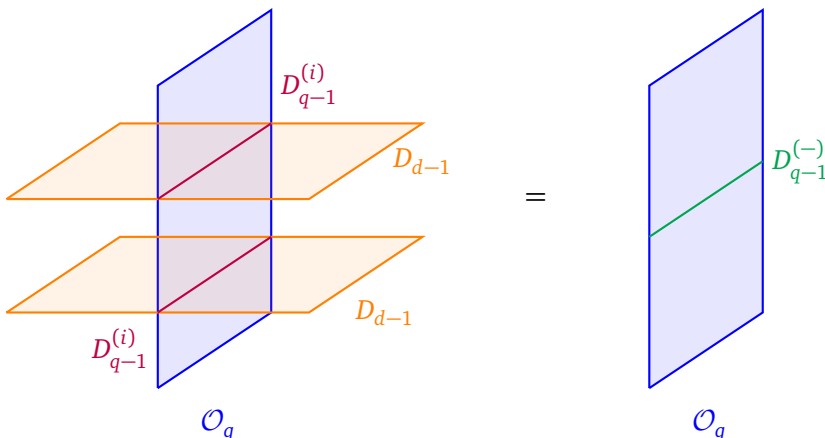

Figure 8: The figure depicts $G^{(0)} = \mathbb{Z}_2$ to $\widetilde{G}^{(0)} = \mathbb{Z}_4$ symmetry fractionalization. $D_{d-1}$ is a topological codimension-1 defect implementing $G^{(0)} = \mathbb{Z}_2$ 0-form symmetry in the bulk and $D_{q-1}^{(i)}$ is a topological defect arising at the intersection of $D_{d-1}$ with a $q$-dimensional operator $\mathcal{O}_q$. The topological defect $D_{q-1}^{(i)}$ induces a $\widetilde{G}^{(0)} = \mathbb{Z}_4$ symmetry on $\mathcal{O}_q$ because as shown in the figure its fusion with itself leaves behind a topological operator $D_{q-1}^{(-)}$ living on the worldvolume of $\mathcal{O}_q$, which in turn generates a $\mathbb{Z}_2$ localized 0-form symmetry of $\mathcal{O}_q$: $D_{q-1}^{(-)} \otimes D_{q-1}^{(-)} = D_{q-1}^{(\mathrm{id})}$.

One can construct in $\mathfrak{T}$ a topological condensation surface operator $D_2^{(\mathbb{Z}_2)}$ by gauging the $\mathbb{Z}_2^{(d-2)}$ symmetry on a surface in spacetime. Then the $G^{(0)} = \mathbb{Z}_2^{(0)}$ 0-form symmetry fractionalizes to a $\widetilde{G}^{(0)} = \mathbb{Z}_4^{(0)}$ on this surface operator $D_2^{(\mathbb{Z}_2)}$, and thus $D_2^{(\mathbb{Z}_2)}$ furnishes such a non-group cohomology 2-charge.

This was shown in section 2.5 of [47] for $d = 3$, but the same argument extends to general $d$.

### 2.3.3 Example: Non-invertible symmetry fractionalization

Generalizing the above story, the physical structure of a general $q$-charge can be understood as the phenomenon of the bulk $G^{(0)}$ 0-form symmetry fractionalizing to a non-invertible induced symmetry on the world-volume of an irreducible multiplet of $q$-dimensional operators furnishing the $q$-charge. When the irreducible multiplet contains a single $q$-dimensional operator $\mathcal{O}_q$, the non-invertible induced symmetry on $\mathcal{O}_q$ is described by the symmetry $(q-1)$ category [17] $\mathcal{C}^{(q-1)}(\mathcal{O}_q)$ discussed around (56).

Below we provide a simple example exhibiting non-invertible symmetry fractionalization, where a $\mathbb{Z}_2^{(0)}$ 0-form symmetry fractionalizes on a surface defect $\mathcal{O}_2$ to a non-invertible induced symmetry described by the Ising fusion category.

---

**Example 2.6: Symmetry fractionalization to Ising category**

Let us conclude this section by providing an illustrative example of non-invertible symmetry fractionalization. This is in fact the simplest example of non-invertible symmetry fractionalization. It is furnished by a surface operator $\mathcal{O}_2$ in a $d$-dimensional QFT $\mathfrak{T}$ carrying a

$$G^{(0)} = \mathbb{Z}_2 \,, \tag{66}$$

---

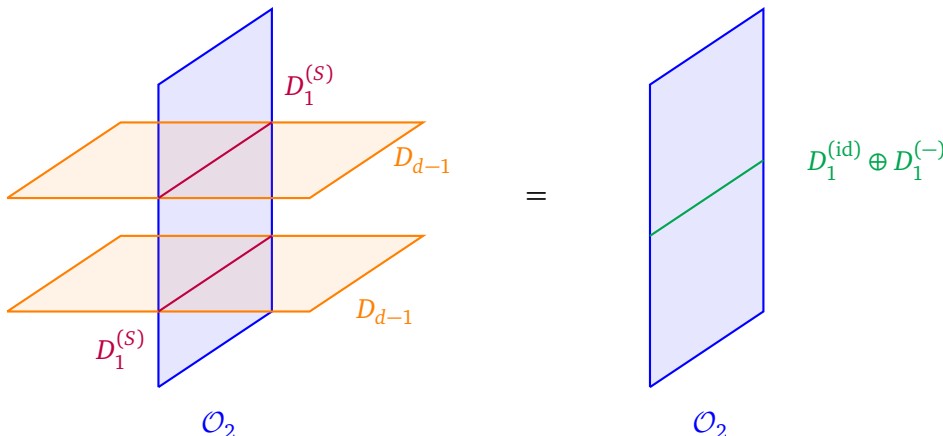

Figure 9: The figure depicts $G^{(0)} = \mathbb{Z}_2$ to $\mathcal{C}$ = Ising non-invertible symmetry fractionalization. $D_{d-1}$ is a topological codimension-1 defect implementing $G^{(0)} = \mathbb{Z}_2$ 0-form symmetry in the bulk and $D_1^{(S)}$ is a topological defect arising at the intersection of $D_{d-1}$ with a surface operator $\mathcal{O}_2$. The topological defect $D_1^{(S)}$ induces Ising symmetry on $\mathcal{O}_2$ because as shown in the figure its fusion with itself leaves behind a topological line operator $D_1^{(\text{id})} \oplus D_1^{(-)}$ living on the world-volume of $\mathcal{O}_2$, where $D_1^{(\text{id})}$ is the identity line on $\mathcal{O}_2$ and $D_1^{(-)}$ generates a $\mathbb{Z}_2$ localized 0-form symmetry of $\mathcal{O}_2$: $D_1^{(-)} \otimes D_1^{(-)} = D_1^{(\text{id})}$.

0-form symmetry generated by a topological operator $D_{d-1}$. The surface operator $\mathcal{O}_2$ has additionally a localized $\mathbb{Z}_2$ 0-form symmetry, generated by a topological line operator $D_1^{(-)}$ living in the worldvolume of $\mathcal{O}_2$. On the other hand, there is an induced localized symmetry

$$D_1^{(S)}, \tag{67}$$

arising at the junction of $\mathcal{O}_2$ and $D_{d-1}$. The fusion of $D_1^{(S)}$ with itself is

$$D_1^{(S)} \otimes D_1^{(S)} = D_1^{(\text{id})} \oplus D_1^{(-)}, \tag{68}$$

where $D_1^{(\text{id})}$ is the identity line operator on $\mathcal{O}_2$. See figure 9. Since this is a non-invertible fusion rule, this means that the bulk $G^{(0)} = \mathbb{Z}_2$ 0-form symmetry fractionalizes to a non-invertible symmetry on $\mathcal{O}_2$. In fact, the non-invertible symmetry can be recognized as the well-known **Ising symmetry** generated by the Ising fusion category, which is discussed in more detail below.

Mathematically, the 2-charge carried by $\mathcal{O}_2$ is described by a 1d 3-representation corresponding to a $\mathbb{Z}_2$-graded fusion category whose underlying non-graded fusion category is the Ising fusion category. This fusion category has three simple objects

$$\left\{ D_1^{(\text{id})}, \ D_1^{(-)}, \ D_1^{(S)} \right\}, \tag{69}$$

along with fusion rules

$$D_1^{(-)} \otimes D_1^{(-)} = D_1^{(\text{id})},$$
$$D_1^{(-)} \otimes D_1^{(S)} = D_1^{(S)} \otimes D_1^{(-)} = D_1^{(S)}, \tag{70}$$
$$D_1^{(S)} \otimes D_1^{(S)} = D_1^{(\text{id})} \oplus D_1^{(-)}.$$

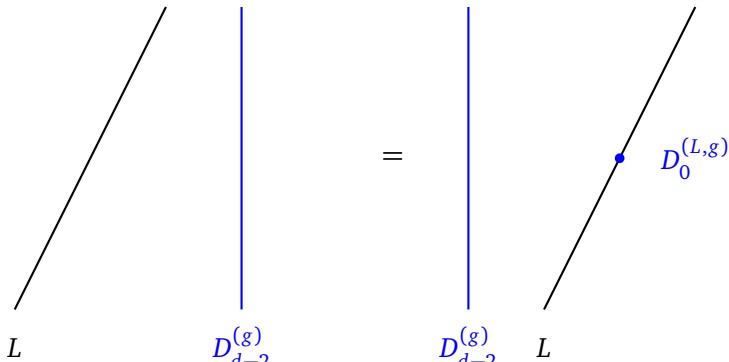

Figure 10: Topological local operators $D_0^{(L,g)}$ living on a line $L$ obtained after moving the 1-form symmetry generator $D_{d-2}^{(g)}$ across $L$.

It is converted into a $\mathbb{Z}_2$-graded fusion category by assigning $\{D_1^{(\mathrm{id})}, D_1^{(-)}\}$ to the trivial grade and $D_1^{(S)}$ to the non-trivial grade.

It would be interesting to find a $\mathbb{Z}_2^{(0)}$ symmetric QFT $\mathfrak{T}$ carrying this 2-charge, which is a problem that is left for future work. The authors think that such 2-charges could be produced by coupling 2d Ising CFT to $\mathfrak{T}$ judiciously, probably in a similar way to the construction of theta topological defects discussed in [26, 46, 47], but now extending the construction to include non-topological defects.

# 3 Generalized charges for 1-form symmetries

In this section, we discuss generalized charges of a 1-form symmetry group $G^{(1)}$. As for 0-form symmetries the simplest instance is the case of 1-charges, upon which the symmetry acts simply as representations. However, we will see again that higher $q$-charges, i.e. $q$-dimensional operators upon which the 1-form symmetry acts, are associated to higher-representations. Let us emphasize that these are not higher-representations of the group $G^{(1)}$, but rather higher-representations of the 2-group $\mathbb{G}_{G^{(1)}}^{(2)}$ associated to the 1-form group $G^{(1)}$. We will denote the generators of the 1-form symmetry group by topological codimension-2 operators

$$D_{d-2}^{(g)}, \qquad g \in G^{(1)}. \tag{71}$$

## 3.1 1-charges

The action of a 1-form symmetry on line operators is similar to the action of a 0-form symmetry on local operators [5]. We can move a codimension-2 topological operator labeled by $g \in G^{(1)}$ across a line operator $L$. In the process, we may generate a topological local operator $D_0^{(L,g)}$ living on $L$. See figure 10. The consistency condition analogous to (16) is

$$D_0^{(L,g)} \otimes D_0^{(L,g')} = D_0^{(L,gg')}. \tag{72}$$

Since $L$ is assumed to be simple, the operators $D_0^{(L,g)}$ can be identified with elements of $\mathbb{C}^\times$, and then the above condition means that $L$ corresponds to an irreducible representation (or character) of the abelian group $G^{(1)}$. This is simply the special case $p = 1$ of statement 1.1.

In fact, mathematically, representations of the 1-form symmetry group $G^{(1)}$ are the same as 2-representations of the 2-group $\mathbb{G}_{G^{(1)}}^{(2)}$ based on the 1-form group $G^{(1)}$. What we mean by

$\mathbb{G}^{(2)}_{G^{(1)}}$ is simply the 2-group which is comprised of a trivial 0-form symmetry and said 1-form symmetry $G^{(1)}$.

Thus, we recover the $p = q = 1$ version of the statement 1.3:

> **Statement 3.1: 1-charges for 1-form symmetries**
>
> 1-charges of a $G^{(1)}$ 1-form symmetry are 2-representations of the associated 2-group $\mathbb{G}^{(2)}_{G^{(1)}}$.

### 3.2 2-charges

In this subsection, we want to understand how a simple surface operator $\mathcal{O}_2$ interacts with a $G^{(1)}$ 1-form symmetry.

**Localized and induced symmetries.** Again, there are two types of symmetries: localized symmetries which only exist on the surface operator, and those that arise from intersections with bulk topological operators, the induced symmetries.

First of all, recall from the discussion of section 2.3.2 that a simple surface operator $\mathcal{O}_2$ can carry localized symmetries generated by topological line operators living on its worldvolume, and the localized symmetry may in general be non-invertible.

Mathematically, the localized symmetries of $\mathcal{O}_2$ are captured by a fusion category $\mathcal{C}$. The different localized symmetries correspond to different simple objects $X \in \mathcal{C}$, and we label the corresponding topological line operators by

$$D_1^{(X)}, \quad X \in \mathcal{C}. \tag{73}$$

The invertible part of localized symmetries described by $\mathcal{C}$ will play a special role in the discussion that follows. This is described by a group $H_{\mathcal{C}}$ which we refer to as the 0-form symmetry localized on $\mathcal{O}_2$. We label the corresponding topological line operators as $D_1^{(h)}$ for $h \in H_{\mathcal{C}}$.

Additionally we have 1-form symmetries induced on $\mathcal{O}_2$ by the bulk 1-form symmetry which are generated by topological local operators $D_0^{(g)}$ arising at the junctions of $\mathcal{O}_2$ with $D_{d-2}^{(g)}$.

**Induced symmetries sourcing localized symmetries.** Just as for the case of 0-form symmetries discussed in section 2.3.2, the induced symmetries may interact non-trivially with the localized symmetries. There are two possible interactions at play here, which we discuss in turn.

The first interaction is that the junction topological local operator $D_0^{(g)}$ may be attached to a topological line operator $D_1^{(X_g)}$ generating a localized symmetry of $\mathcal{O}_2$. In other words, the induced symmetry generated by $D_0^{(g)}$ sources (a background of) the localized symmetry generated by $D_1^{(X_g)}$. See figure 11.

The composition rule

$$D_{d-2}^{(g)} \otimes D_{d-2}^{(g')} = D_{d-2}^{(gg')}, \tag{74}$$

of 1-form symmetries needs to be obeyed by the sourced localized symmetries

$$D_1^{(X_g)} \otimes D_1^{(X_{g'})} = D_1^{(X_{gg'})}. \tag{75}$$

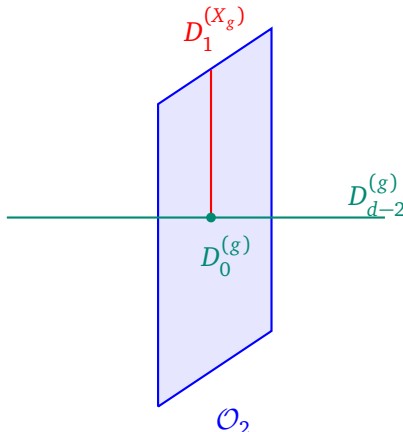

Figure 11: $D_0^{(g)}$ are topological local operators inducing $G^{(1)}$ 1-form symmetry on the surface operator $\mathcal{O}_2$. These operators arise at the junction of $\mathcal{O}_2$ with the bulk topological operators $D_{d-2}^{(g)}$. Additionally, $D_0^{(g)}$ may be attached to a topological line operator $D_1^{(X_g)} \in \mathcal{C}$ generating a localized symmetry of $\mathcal{O}_2$.

As a consequence of this, only the invertible localized 0-form symmetries described by group $H_{\mathcal{C}}$ can be sourced by the induced 1-form symmetries. We thus have a homomorphism

$$\tau : \ G^{(1)} \to H_{\mathcal{C}}, \tag{76}$$

describing the localized 0-form symmetry sourced by each induced 1-form symmetry element and we can write

$$D_1^{(X_g)} = D_1^{\left(\tau(g)\right)}. \tag{77}$$

This results in the following non-trivial constraint on the possible background fields

$$\delta A_1 = \tau \left( B_2|_{\mathcal{O}_2} \right), \tag{78}$$

where $A_1$ is the background field for localized $H_{\mathcal{C}}$ 0-form symmetry living on the world-volume of $\mathcal{O}_2$, and $B_2|_{\mathcal{O}_2}$ is the restriction onto the world-volume of $\mathcal{O}_2$ of the background field $B_2$ for the $G^{(1)}$ 1-form symmetry living in the $d$-dimensional bulk.

Later, in example 3.2, we will describe a surface operator (called $\mathcal{O}_2'$ there) in the pure 4d $O(4N)$ gauge theory exhibiting $\mathbb{Z}_2$ 0-form localized symmetry which is sourced by the $\mathbb{Z}_2$ 1-form symmetry induced from the 4d bulk.

**Action of localized symmetries on induced symmetries: Mixed 't hooft anomaly.** The second interaction between localized and induced symmetries is that the former can act on the latter. In other words, **induced symmetries correspond to 0-charges for (possibly non-invertible) localized symmetries**. A detailed discussion of the structure of generalized charges for non-invertible symmetries is the subject of Part II in this series [1].

The action is implemented by moving a topological line operator $D_1^{(X)}$ past a junction topological local operator $D_0^{(g)}$ as shown in figure 12. After this move, the junction of $\mathcal{O}_2$ and $D_{d-2}^{(g)}$ sources localized symmetry

$$D_1^{(X)} \otimes D_1^{(\tau(g))} \otimes D_1^{(X^*)}. \tag{79}$$

But we have already established that the junction can only source $D_1^{(\tau(g))}$, which implies that the line (79) must equal $D_1^{(\tau(g))}$. Restricting to the invertible part $D_1^{(X)} = D_1^{(h)}$ for $h \in H_{\mathcal{C}}$, we

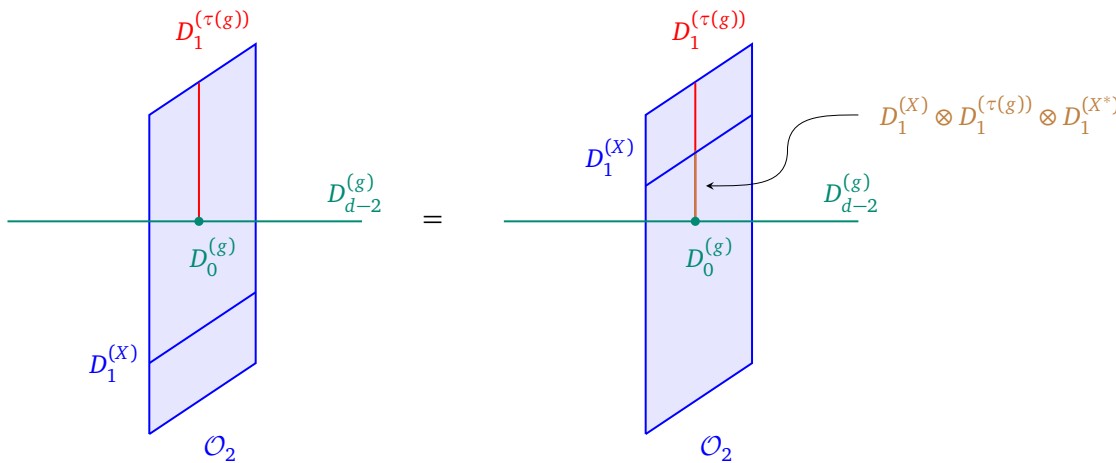

Figure 12: Moving an arbitrary topological line operator $D_1^{(X)}$ living on the world-volume of $\mathcal{O}_2$ past a junction topological local operator $D_0^{(g)}$.

learn that the image of the homomorphism (76) is contained in the center $Z(H_\mathcal{C})$ of $H_\mathcal{C}$. That is, induced 1-form symmetries can only source localized symmetries lying in $Z(H_\mathcal{C})$.

The action of the $H_\mathcal{C}$ localized 0-form symmetry is

$$h \in H_\mathcal{C}: \ D_0^{(g)} \to \phi_g(h) D_0^{(g)}, \tag{80}$$

where $\phi_g(h) \in \mathbb{C}^\times$ such that

$$\phi_g(h)\phi_g(h') = \phi_g(hh'), \qquad \phi_g(1) = 1. \tag{81}$$

This means that the junction topological local operator $D_0^{(g)}$ furnishes a 1-dimensional representation of $H_\mathcal{C}$. Thus we have an homomorphism

$$\phi: \ G^{(1)} \to \chi(H_\mathcal{C}), \tag{82}$$

where $\chi(H_\mathcal{C})$ is the *character group*, namely the group formed by 1-dimensional representations, of $H_\mathcal{C}$.

The action of $H_\mathcal{C}$ on induced 1-form symmetries can be viewed as a mixed 't Hooft anomaly between the localized 0-form and the induced 1-form symmetries of the form

$$\mathcal{A}_3 = \phi_{B_2|_{\mathcal{O}_2}}(A_1), \tag{83}$$

where the notation for background fields has been discussed above. More concretely, $\mathcal{A}_3$ is a $\mathbb{C}^\times$ valued 3-cocycle whose explicit simplicial form is

$$\mathcal{A}_3(v_0, v_1, v_2, v_3) = \phi_{B_2|_{\mathcal{O}_2}(v_0, v_1, v_2)}(A_1(v_2, v_3)) \in \mathbb{C}^\times, \tag{84}$$

where $v_i$ are vertices in a simplicial decomposition, $B_2|_{\mathcal{O}_2}(v_0, v_1, v_2) \in G^{(1)}$ and $A_1(v_2, v_3) \in H_\mathcal{C}$.

Note that, for consistency we must demand that

$$\delta A_3 = 0, \tag{85}$$

which is a non-trivial condition to satisfy due to the non-closure condition (78).

Later, in example 3.2, we will describe a surface operator (called $\mathcal{O}_2$ there) in the pure 4d $O(4N)$ gauge theory exhibiting $\mathbb{Z}_2$ 0-form localized symmetry under which the topological local operator generating $\mathbb{Z}_2$ 1-form symmetry induced from the 4d bulk is non-trivially charged.

**'t Hooft anomaly for induced 1-form symmetry.** Finally, we have a constraint that the composition of induced 1-form symmetries is consistent with the composition (74) of bulk 1-form symmetries, implying that we must have

$$D_0^{(g)} \otimes D_0^{(g')} = \alpha(g, g') D_0^{(gg')},$$ (86)

where $\alpha$ is a $\mathbb{C}^\times$ valued 2-cochain on $G^{(1)}$.

Moreover, the associativity of (74) imposes the condition that $\alpha$ is a 2-cocycle. In fact, using the freedom to rescale topological local operators $D_0^{(g)}$, only the cohomology class

$$[\alpha] \in H^2(G^{(1)}, \mathbb{C}^\times),$$ (87)

distinguishes different 2-charges.

Physically, the class $[\alpha]$ describes a pure 't Hooft anomaly for the $G^{(1)}$ induced symmetry taking the form

$$\mathcal{A}_3 = \exp\left(2\pi i \int \text{Bock}(B_2|_{\mathcal{O}_2})\right),$$ (88)

where the Bockstein is taken with respect to the short exact sequence

$$0 \to \mathbb{C}^\times \to \widetilde{G}^{(1)} \to G^{(1)} \to 0,$$ (89)

specified by the extension class $[\alpha]$.

**Mathematical structure.** All of the above information describing a 2-charge of 1-form symmetry can be neatly encapsulated using category theory. First of all, as we have already been using in the above physical description, the localized symmetries are described by a fusion category $\mathcal{C}$. The interactions of localized and induced symmetries, along with the pure 't Hooft anomaly of induced symmetries is mathematically encapsulated in the information of a braided monoidal functor

$$\text{Vec}(G^{(1)}) \to \mathcal{Z}(\mathcal{C}),$$ (90)

where $\text{Vec}(G^{(1)})$ is the braided fusion category obtained by giving a trivial braiding to the fusion category formed by $G^{(1)}$-graded vector spaces and $\mathcal{Z}(\mathcal{C})$ is the modular (in particular braided) fusion category formed by the Drinfeld center of $\mathcal{C}$.

Let us expand on how this mathematical structure encodes all of the physical information discussed above. As we will argue in Part II [1], we have the following general statement.

> **Statement 3.2: 0-charges of a non-invertible categorical symmetry**
>
> The 0-charges of a possibly non-invertible symmetry described by a fusion category $\mathcal{C}$ are objects of its Drinfeld center $\mathcal{Z}(\mathcal{C})$.

Then, the functor (90) assigns to every 1-form symmetry element $g \in G^{(1)}$ a 0-charge for the localized symmetry $\mathcal{C}$ on $\mathcal{O}_2$. More concretely, an object of Drinfeld center $\mathcal{Z}(\mathcal{C})$ can be expressed as

$$(X, \beta),$$ (91)

where $X$ is an object in $\mathcal{C}$ and $\beta$ is a collection of morphisms in $\mathcal{C}$ involving $X$. The functor (90) thus assigns to $g \in G^{(1)}$ a simple object $(X_g = \tau(g), \beta_g) \in \mathcal{Z}(\mathcal{C})$ where the simple object $X_g = \tau(g) \in Z(H_{\mathcal{C}})$ describes the localized symmetry sourced by the corresponding induced 1-form symmetry and the morphisms $\beta_g$ encode the action of localized symmetries on induced

symmetries. This encoding will be described in Part II [1]. Finally, the fact that the functor is monoidal encodes the condition (86) along with the characterization (87).

Such functors capture precisely 1-dimensional 3-representations of the 2-group $\mathbb{G}^{(2)}_{G^{(1)}}$ based on the 1-form group $G^{(1)}$, and general 3-representations are direct sums of these 1-dimensional ones. Thus, we recover the $p = 1, q = 2$ piece of statement 1.3:

---

**Statement 3.3: 2-charges for 1-form symmetries**

2-charges of a $G^{(1)}$ 1-form symmetry are 3-representations of the associated 2-group $\mathbb{G}^{(2)}_{G^{(1)}}$.

---

Let us describe some simple examples of non-trivial 2-charges:

---

**Example 3.1: 2-charges for $G^{(1)} = \mathbb{Z}_2$**

As an illustration, let us enumerate all the possible 2-charges for

$$G^{(1)} = \mathbb{Z}_2 \,, \tag{92}$$

1-form symmetry that exhibit

$$\mathcal{C} = \mathsf{Vec}_{\mathbb{Z}_2} \,, \tag{93}$$

localized symmetry, which corresponds to a

$$G^{(0)}_{\mathcal{O}_2} = \mathbb{Z}_2 \,, \tag{94}$$

non-anomalous 0-form symmetry localized on the corresponding surface operator $\mathcal{O}_2$.

First of all, the induced 1-form symmetry cannot carry a pure 't Hooft anomaly because

$$H^2(\mathbb{Z}_2, \mathbb{C}^\times) = 0 \,. \tag{95}$$

Thus, we have the following possible 2-charges:

1. There is no interaction between the localized and induced symmetries.

2. The generator of the induced $\mathbb{Z}_2$ 1-form symmetry is charged under the localized symmetry (94).

   This corresponds to a 't Hooft anomaly

   $$\mathcal{A}_3 = \exp\left( \int A_1 \cup B_2|_{\mathcal{O}_2} \right). \tag{96}$$

3. The generator of the induced $\mathbb{Z}_2$ 1-form symmetry is in the twisted sector for the generator of the localized symmetry (94). In other words, the induced symmetry sources the localized symmetry.

   In terms of background fields, we have the relationship

   $$\delta A_1 = B_2|_{\mathcal{O}_2} \,. \tag{97}$$

---

Note that the generator of the induced $\mathbb{Z}_2$ 1-form symmetry cannot be both charged and be in the twisted sector at the same time, because in such a situation the relationship (97) would force the mixed 't Hooft anomaly (96) to be non-closed

$$\delta \mathcal{A}_3 = \exp\left( \int B_2|_{\mathcal{O}_2} \cup B_2|_{\mathcal{O}_2} \right) \neq 0, \tag{98}$$

which is a contradiction.

**Categorical formulation.** We can also recover the above three possibilities using the more mathematical approach outlined above. Mathematically, we want to enumerate braided monoidal functors from the braided fusion category $\mathsf{Vec}_{G^{(1)}=\mathbb{Z}_2}$ (with trivial braiding) to the modular tensor category $\mathcal{Z}(\mathcal{C} = \mathsf{Vec}_{\mathbb{Z}_2})$. The latter can be recognized as the category describing topological line defects of the 3d $\mathbb{Z}_2$ Dijkgraaf-Witten gauge theory, or in other words the 2+1d toric code. In other words, we are enumerating different ways of choosing a non-anomalous $\mathbb{Z}_2$ 1-form symmetry of the above 3d TQFT. The simple topological line operators of the 3d TQFT are

$$\{1, e, m, \psi\}, \tag{99}$$

with fusions $e \otimes e = m \otimes m = \psi \otimes \psi = 1$ and $\psi = e \otimes m$, and spins $\theta(e) = \theta(m) = 1$ and $\theta(\psi) = -1$. There are precisely three choices for a non-anomalous $\mathbb{Z}_2$ 1-form symmetry:

1. Choose the identity line 1 as the generator of the $\mathbb{Z}_2$ 1-form symmetry. This corresponds to the 2-charge in which there is no interaction between the induced and localized symmetries.

2. Choose the "electric" line $e$ as the generator of the $\mathbb{Z}_2$ 1-form symmetry. This corresponds to the 2-charge in which the induced symmetry is charged under localized symmetry.

3. Choose the "magnetic" line $m$ as the generator of the $\mathbb{Z}_2$ 1-form symmetry. This corresponds to the 2-charge in which the induced symmetry sources the localized symmetry.

Note that we cannot choose the "dyonic"/"fermionic" line $\psi$ as the generator of $\mathbb{Z}_2$ 1-form symmetry, because the $\psi$ line is a fermion (recall $\theta(\psi) = -1$) and hence generates a $\mathbb{Z}_2$ 1-form symmetry with a non-trivial 't Hooft anomaly. This corresponds to the fact that one cannot have a 2-charge in which induced symmetry is both charged under the localized symmetry and also sources the localized symmetry.

**Pairing of 2-charges.** Note that one can obtain a surface operator $\mathcal{O}_2'$ carrying a 2-charge having property (97) from a surface operator $\mathcal{O}_2$ carrying 2-charge having property (96) by gauging the localized $\mathbb{Z}_2$ 0-form symmetry of $\mathcal{O}_2$. After the gauging procedure we obtain a dual $\mathbb{Z}_2$ 0-form localized symmetry on $\mathcal{O}_2'$. The local operator generating the induced $\mathbb{Z}_2$ 1-form symmetry is charged under the original localized $\mathbb{Z}_2$, and so becomes a twisted sector local operator for the dual $\mathbb{Z}_2$. Similarly, one can obtain $\mathcal{O}_2$ from $\mathcal{O}_2'$ by gauging the localized $\mathbb{Z}_2$ 0-form symmetry of $\mathcal{O}_2'$.

Thus, if a theory admits a surface operator carrying 2-charge corresponding to the electric line $e$, then it also must admit a surface operator carrying 2-charge corresponding to the magnetic line $m$.

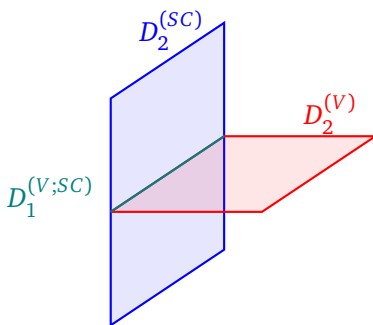

Figure 13: The topological line $D_1^{(V;SC)}$ arising at the end of the topological surface $D_2^{(V)}$ along the topological surface $D_2^{(SC)}$ in the 4d pure $O(4N)$ gauge theory.

Below we describe a concrete field theory which realizes the above discussed 2-charges.

---

**Example 3.2: 4d $O(4N)$ gauge theory**

The two non-trivial 2-charges exhibiting properties (97) and (96) are realized in 4d pure $O(4N)$ gauge theory. This can be easily seen if we begin with the 4d pure $\text{Pin}^+(4N)$ gauge theory, which as discussed in [26] has topological surface operators described by 2-representations of a split 2-group. We will only use two surface operators $D_2^{(SC)}$ and $D_2^{(V)}$ having fusions

$$
\begin{aligned}
D_2^{(V)} \otimes D_2^{(V)} &= D_2^{(\text{id})}, \\
D_2^{(V)} \otimes D_2^{(SC)} &= D_2^{(SC)},
\end{aligned}
\tag{100}
$$

with $D_2^{(SC)}$ having non-invertible fusion with itself. The first fusion implies that $D_2^{(V)}$ generates a $\mathbb{Z}_2$ 1-form symmetry, which can be identified as the center 1-form symmetry of the $\text{Pin}^+(4N)$ theory. On the other hand, the second fusion implies that there is a line operator $D_1^{(V;SC)}$ living at an end of $D_2^{(V)}$ along $D_2^{(SC)}$. See figure 13. This line operator is $\mathbb{Z}_2$ valued: due to the first fusion rule, its square must be a line operator living on the world-volume of $D_2^{(SC)}$, but the only such line operator is the identity one.

Gauging the $\mathbb{Z}_2$ 1-form symmetry generated by $D_2^{(V)}$ leads to the $O(4N)$ theory. The surface operator furnishing the desired 2-charge is

$$
\mathcal{O}_2 = D_2^{(SC)},
\tag{101}
$$

or more precisely the image in the $O(4N)$ theory of the operator $D_2^{(SC)}$ of the $\text{Pin}^+(4N)$ theory. The relevant 1-form symmetry

$$
G^{(1)} = \mathbb{Z}_2,
\tag{102}
$$

is the dual 1-form symmetry arising after the above gauging, which can be identified as the magnetic 1-form symmetry of the $O(4N)$ theory.

The line operator $D_1^{(V;SC)}$ becomes a line operator living on the world-volume of $D_2^{(SC)}$, thus generating a

$$
G_{\mathcal{O}_2}^{(0)} = \mathbb{Z}_2,
\tag{103}
$$

localized symmetry of it. Additionally, the generator $D_1^{(V;SC)}$ of this localized symmetry has to be charged under the $\mathbb{Z}_2$ 1-form symmetry induced on $\mathcal{O}_2$ by the bulk magnetic

---

1-form symmetry, because before the gauging the line operator $D_1^{(V;SC)}$ lied at the end of the topological operator $D_2^{(V)}$ being gauged. This means that we have a 't Hooft anomaly (96) and hence the proposed surface operator $\mathcal{O}_2$ indeed furnishes the desired non-trivial 2-charge.

A surface operator $\mathcal{O}_2'$ furnishing the other non-trivial 2-charge is simply obtained from $\mathcal{O}_2$ by gauging its $\mathbb{Z}_2$ localized symmetry along its whole world-volume. As explained above, $\mathcal{O}_2'$ then exhibits (97).

## 3.3 Higher-charges

Continuing in the above fashion, one may study $q$-charges for $q \geq 3$. The interesting physical phenomenon that opens up here is the possibility of fractionalization of 1-form symmetry, i.e. the induced 1-form symmetry on a $q$-dimensional operator $\mathcal{O}_q$ may be a larger group $\widetilde{G}^{(1)}$, or may be a larger higher-group, or the induced symmetry may actually be non-invertible. Mathematically, such $q$-charges are expected to form $(q+1)$-representations $\rho^{(q+1)}$ of the 2-group $\mathbb{G}_{G^{(1)}}^{(2)}$ associated to the 1-form symmetry group $G^{(1)}$.

**1-form symmetry fractionalization in special types of 3-charges.** Let us discuss a special class of 3-charges which exhibit both invertible and non-invertible 1-form symmetry fractionalization. An irreducible 3-charge in this class exhibits the following physical properties:

1. The localized symmetries on a 3-dimensional operator $\mathcal{O}_3$ furnishing the 3-charge are captured essentially by topological line operators living on $\mathcal{O}_3$. These line operators can not only have non-invertible fusion rules, but also can braid non-trivially with each other. These line operators and their properties are encoded mathematically in the structure of a braided fusion category $\mathcal{B}$.

2. The world-volume of $\mathcal{O}_3$ also contains topological surface operators, but in this special class of 3-charges, all such surface operators can be constructed by gauging/condensing the topological line operators in $\mathcal{B}$ along two-dimension sub-world-volumes of $\mathcal{O}_3$. Mathematically, the topological surfaces and lines combine together to form a fusion 2-category denoted by

$$\mathrm{Mod}(\mathcal{B}), \tag{104}$$

whose objects are module categories of $\mathcal{B}$.

3. The induced symmetries arise at codimension-2 (i.e. line-like) junctions of the bulk $G^{(1)}$ 1-form symmetry generators and $\mathcal{O}_3$. There can be various types of topological line operators arising at such junctions, which generate the induced symmetries. Combining them with line operators in $\mathcal{B}$, we obtain the mathematical structure of a $G^{(1)}$-graded braided fusion category $\mathcal{B}_{G^{(1)}}$.

4. Physically, in such a 3-charge, the induced symmetries do *not* source localized symmetries, but the generators of induced symmetries can be charged under localized symmetries. This information about the charge is encoded mathematically in the braiding of an arbitrary object of the graded category $\mathcal{B}_{G^{(1)}}$ with an object of the trivially graded part $\mathcal{B} \subseteq \mathcal{B}_{G^{(1)}}$.

A generic choice of $\mathcal{B}_{G^{(1)}}$ corresponds to a non-invertible fractionalization of $G^{(1)}$ 1-form symmetry, quite similar to the non-invertible fractionalization of 0-form symmetry discussed in section 2.3.2.

> **Example 3.3: Invertible and non-invertible 1-form symmetry fractionalization**
>
> Let us provide examples of $\mathcal{B}_{G^{(1)}}$ corresponding to both invertible and non-invertible symmetry fractionalization for
>
> $$G^{(1)} = \mathbb{Z}_2 \,. \tag{105}$$
>
> For invertible symmetry fractionalization, take
>
> $$\mathcal{B}_{G^{(1)}} = \mathsf{Vec}_{\mathbb{Z}_4} \,, \tag{106}$$
>
> with trivial braiding, and grading specified by surjective homomorphism $\mathbb{Z}_4 \to \mathbb{Z}_2$. A 3-dimensional operator $\mathcal{O}_3$ furnishing such a 3-charge carries a
>
> $$G^{(1)}_{\mathcal{O}_3} = \mathbb{Z}_2 \,, \tag{107}$$
>
> 1-form localized symmetry, which is extended to a total of $G^{(1)}_{\text{ind-loc}} = \mathbb{Z}_4$ 1-form symmetry by 1-form symmetries induced on $\mathcal{O}_3$ from the bulk $G^{(1)} = \mathbb{Z}_2$ 1-form symmetry:
>
> $$0 \longrightarrow G^{(1)}_{\mathcal{O}_3} = \mathbb{Z}_2 \longrightarrow G^{(1)}_{\text{ind-loc}} = \mathbb{Z}_4 \longrightarrow G^{(1)} = \mathbb{Z}_2 \longrightarrow 0 \,. \tag{108}$$
>
> In other words, the bulk $G^{(1)} = \mathbb{Z}_2$ 1-form symmetry is fractionalized to $G^{(1)}_{\text{ind-loc}} = \mathbb{Z}_4$ 1-form symmetry on the worldvolume of $\mathcal{O}_3$.
>
> For non-invertible symmetry fractionalization, take $\mathcal{B}_{G^{(1)}}$ to be Ising modular fusion category, and grading that assigns trivial grade to $\{D_1^{(\text{id})}, D_1^{(-)}\}$ and non-trivial grade to $D_1^{(S)}$. See example 2.6 for details on notation. The bulk $G^{(1)} = \mathbb{Z}_2$ 1-form symmetry is now fractionalized to non-invertible (1-form) symmetry on the world-volume of $\mathcal{O}_3$ because of the last fusion rule in (70). The localized symmetry is still
>
> $$G^{(1)}_{\mathcal{O}_3} = \mathbb{Z}_2 \,, \tag{109}$$
>
> generated by $D_1^{(-)}$, so we can regard Ising category as a categorical extension of $G^{(1)} = \mathbb{Z}_2$ by $G^{(1)}_{\mathcal{O}_3} = \mathbb{Z}_2$, writing an analog of (108)
>
> $$\text{``}0 \longrightarrow G^{(1)}_{\mathcal{O}_3} = \mathbb{Z}_2 \longrightarrow G^{(1)}_{\text{ind-loc}} = \text{Ising} \longrightarrow G^{(1)} = \mathbb{Z}_2 \longrightarrow 0\text{''} \,. \tag{110}$$

# 4 Non-genuine generalized charges

So far we considered only genuine $q$-charges. As we will discuss now, non-genuine charges arise naturally in field theories and require an extension, to include a higher-categorical structure. The summary of this structure can be found in statement 1.5. In this section, we physically study and verify that the statement is correct for 0-charges of 0-form and 1-form symmetries.

## 4.1 Non-genuine 0-charges of 0-form symmetries

We have discussed above that genuine 0-charges for $G^{(0)}$ 0-form symmetry are representations of $G^{(0)}$. Similarly, genuine 1-charges are 2-representations of $G^{(0)}$. In this subsection, we discuss non-genuine 0-charges going from a genuine 1-charge corresponding to a

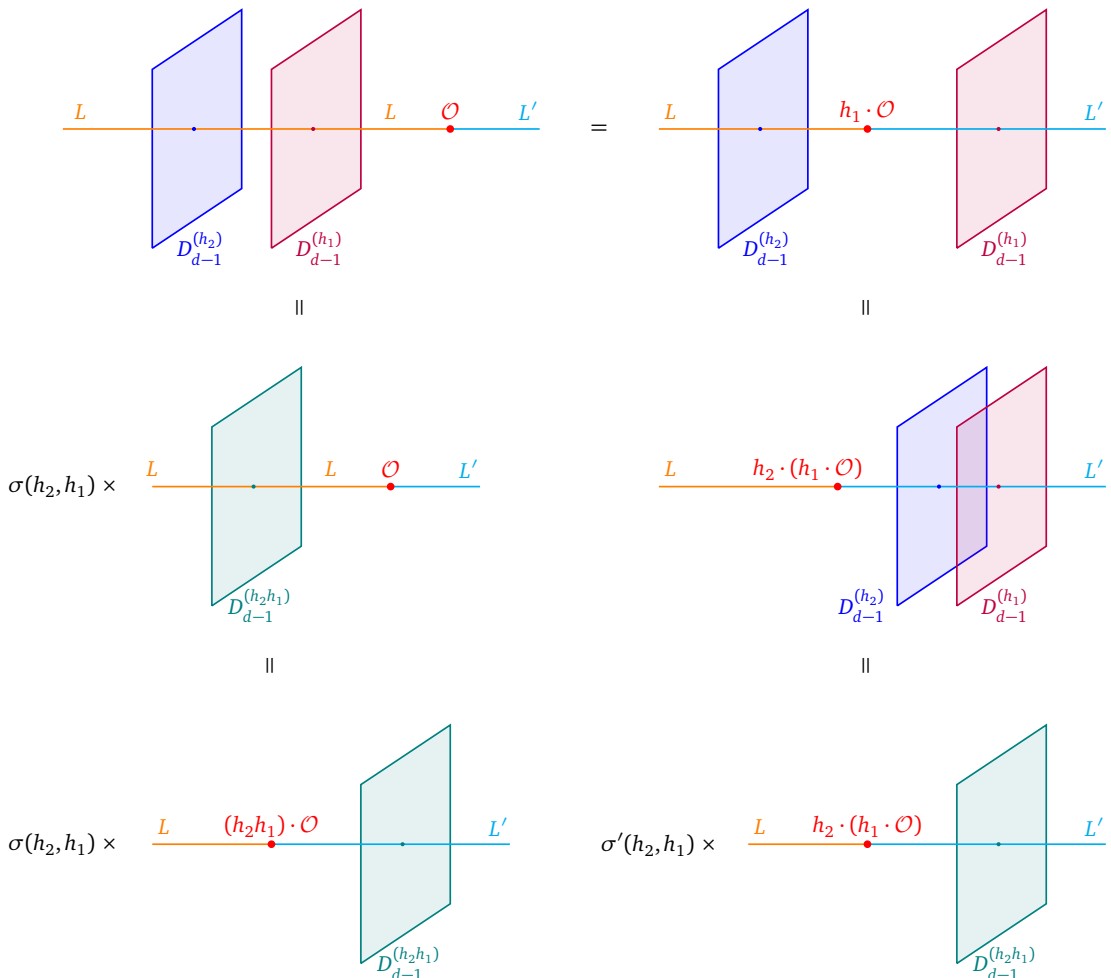

Figure 14: The action of $h_1, h_2 \in H_{LL'}$ on a non-genuine local operator $\mathcal{O}$. Depending on whether we fuse first and then act, or first act and then fuse, we generate two different situations, shown at the bottom-left and bottom-right of the figure. A chain of equalities equates the two, leading to the equation (114).

2-representation $\rho^{(2)}$ to another genuine 1-charge corresponding to a 2-representation $\rho'^{(2)}$. These non-genuine 0-charges are furnished by non-genuine local operators changing a line operator $L$ having 1-charge $\rho^{(2)}$ to a line operator $L'$ having 1-charge $\rho'^{(2)}$:

$$\underline{\quad L \qquad\qquad \mathcal{O} \qquad L' \quad} . \tag{111}$$

Let $\rho^{(2)}$ and $\rho'^{(2)}$ be the following irreducible 2-representations

$$\begin{aligned} \rho^{(2)} &= (H_L, [\sigma]), \\ \rho'^{(2)} &= (H_{L'}, [\sigma']). \end{aligned} \tag{112}$$

Consider a local operator $\mathcal{O}$ changing $L$ to $L'$. The subgroup

$$H_{LL'} := H_L \cap H_{L'} \subseteq G^{(0)}, \tag{113}$$

maps $\mathcal{O}$ to other local operators changing $L$ to $L'$. However, this action of $H_{LL'}$ is not linear, and instead satisfies

$$h_2 \cdot (h_1 \cdot \mathcal{O}) = \frac{\sigma(h_2, h_1)}{\sigma'(h_2, h_1)} (h_2 h_1) \cdot \mathcal{O}, \qquad \forall\, h_1, h_2 \in H_{LL'}, \tag{114}$$

as explained in figure 14.

Because of the factor $\sigma\sigma'^{-1}(h_2, h_1) \in \mathbb{C}^\times$, non-genuine 0-charges from genuine 1-charge $\rho^{(2)}$ to genuine 1-charge $\rho'^{(2)}$ are not linear representations of $H_{LL'}$ in general. Such non-genuine 0-charges are linear representations only if

$$[\sigma\sigma'^{-1}] := [\sigma_{H_{LL'}}][\sigma'_{H_{LL'}}]^{-1} = 1 \in H^2(H_{LL'}, \mathbb{C}^\times), \tag{115}$$

where $[\sigma_{H_{LL'}}]$ and $[\sigma'_{H_{LL'}}]$ are the restrictions of $[\sigma]$ and $[\sigma']$ to $H_{LL'}$, and $1 \in H^2(H_{LL'}, \mathbb{C}^\times)$ is the identity element. On the other hand, the non-genuine 0-charges are not linear representations if

$$[\sigma\sigma'^{-1}] \neq 1 \in H^2(H_{LL'}, \mathbb{C}^\times). \tag{116}$$

In this situation, we say the non-genuine 0-charges are **twisted representations** of $H_{LL'}$ lying in the class $[\sigma\sigma'^{-1}] \in H^2(H_{LL'}, \mathbb{C}^\times)$.

---

**Aside: Difference between twisted and projective representations**

Two solutions of (114) give rise to isomorphic twisted representations if they are related by a basis change on the space of local operators. Note that twisted representations with trivial twist are equivalent to linear representations of $H_{LL'}$. Also note that two non-isomorphic $[\kappa]$-twisted representations may be isomorphic as projective representations in the class $[\kappa]$. The condition for being isomorphic as projective representations is often called as projective equivalence which in addition to basis changes allows the modification of the action of $H_{LL'}$ by a function (not necessarily a homomorphism) $\theta : H_{LL'} \to \mathbb{C}^\times$. For example, non-isomorphic one-dimensional linear representations of $H_{LL'}$ are all isomorphic to the trivial linear representation, when viewed as projective representations.

---

Similarly, we can also consider some other lines in the two multiplets $M_L$ and $M_{L'}$. Non-genuine local operators changing $L_{[g]} \in M_L$ to $L_{[g']} \in M_{L'}$ form twisted representations of $H_{L_{[g]}L'_{[g']}} := H_{L_{[g]}} \cap H_{L'_{[g']}} \subseteq G^{(0)}$.

In fact, mathematically, all these twisted representations combine together to form a 1-morphism in the 2-category 2-Rep($G^{(0)}$) formed by 2-representations of the group $G^{(0)}$. Such 1-morphisms are also referred to as **intertwiners** between the two 2-representations. Indeed, in our physical setup the non-genuine local operators lying between lines $L, L'$ explicitly intertwine the action of 0-form symmetry $G^{(0)}$ on the two line operators. Thus, we have justified the $p = q = 1$ version of statement 1.5.

---

**Statement 4.1: Non-genuine 0-charges of 0-form symmetries**

The possible 0-charges going from a 1-charge described by an object (i.e. a 2-representation) $\rho^{(2)} \in$ 2-Rep($G^{(0)}$) to a 1-charge described by an object $\rho'^{(2)} \in$ 2-Rep($G^{(0)}$), are described by 1-morphisms from the object $\rho^{(2)}$ to the object $\rho'^{(2)}$ in 2-Rep($G^{(0)}$).

---

When $\rho^{(2)}$ and $\rho'^{(2)}$ are both trivial 2-representations, then the intertwiners are the same as representations of $G^{(0)}$. Since the identity line operator necessarily transforms in trivial 2-representation, we hence recover the statement 2.1 regarding genuine 0-charges.

**Example 4.1: Fractional monopole operators**

Consider the example 2.2 of 3d pure $SO(4N)$ gauge theory. As we discussed earlier, the topological line operator $D_1^{(-)}$ generating the $\mathbb{Z}_2^{(1)}$ center 1-form symmetry transforms in a non-trivial 2-representation (38) of $G^{(0)} = \mathbb{Z}_2^m \times \mathbb{Z}_2^o$. Following the analysis above, the non-genuine local operators living at the end of $D_1^{(-)}$ should form twisted instead of linear representations of $G^{(0)} = \mathbb{Z}_2^m \times \mathbb{Z}_2^o$. This means that the actions of $\mathbb{Z}_2^m$ and $\mathbb{Z}_2^o$ on such non-genuine local operators should anti-commute.

We can see this explicitly for special examples of such non-genuine local operators known as fractional gauge monopole operators [67]. In our case, these are local operators that induce monopole configurations for $PSO(4N) = SO(4N)/\mathbb{Z}_2$ on a small sphere $S^2$ surrounding them

$$D_1^{(-)} \quad\rule{2cm}{0.4pt}\quad \bullet \quad S^2 \tag{117}$$

that cannot be lifted to monopole configurations for $SO(4N)$. Such fractional monopole operators can be further divided into two types:

1. The associated monopole configuration for $PSO(4N)$ can be lifted to a monopole configuration for $Ss(4N) = \mathrm{Spin}(4N)/\mathbb{Z}_2^S$ but not to a monopole configuration for $Sc(4N) = \mathrm{Spin}(4N)/\mathbb{Z}_2^C$ or $SO(4N) = \mathrm{Spin}(4N)/\mathbb{Z}_2^V$.

2. The associated monopole configuration for $PSO(4N)$ can be lifted to a monopole configuration for $Sc(4N)$ but not to a monopole configuration for $Ss(4N)$ or $SO(4N)$.

On the other hand, the monopole operators associated to monopole configurations for $PSO(4N)$ that can be lifted to monopole configurations for $SO(4N)$ but not to monopole configurations for $Ss(4N)$ or $Sc(4N)$ are non-fractional monopole operators, which are genuine local operators charged under $\mathbb{Z}_2^m$ 0-form symmetry. Here $\mathbb{Z}_2^S \times \mathbb{Z}_2^C$ is the center of $\mathrm{Spin}(4N)$. The generator of $\mathbb{Z}_2^S$ leaves the spinor representation invariant, but acts non-trivially on the cospinor and vector representations. Similarly, the generator of $\mathbb{Z}_2^C$ leaves the cospinor representation invariant, but acts non-trivially on the spinor and vector representations. Finally, the diagonal $\mathbb{Z}_2$ subgroup is denoted as $\mathbb{Z}_2^V$ whose generator leaves the vector representation invariant, but acts non-trivially on the cospinor and spinor representations.

Now, these two types of fractional monopole operators are exchanged by the outer-automorphism 0-form symmetry $\mathbb{Z}_2^o$. On the other hand, only one of the two types of operators are non-trivially charged under $\mathbb{Z}_2^m$ 0-form symmetry. This is because the two types of fractional monopole operators are interchanged upon taking OPE with non-fractional monopole operators charged under $\mathbb{Z}_2^m$. Thus, fractional monopole operators indeed furnish representations twisted by the non-trivial element of (37) because the actions of $\mathbb{Z}_2^m$ and $\mathbb{Z}_2^o$ anti-commute on these operators.

## 4.2 Non-genuine 0-charges of 1-form symmetries: Absence of screening

In the previous subsection, we saw that there exist 0-charges between two different irreducible 1-charges for a 0-form symmetry. However, the same is not true for the 1-form symmetry. There are no possible 0-charges between two different irreducible 1-charges. This means that there cannot exist non-genuine local operators between two line operators carrying two different

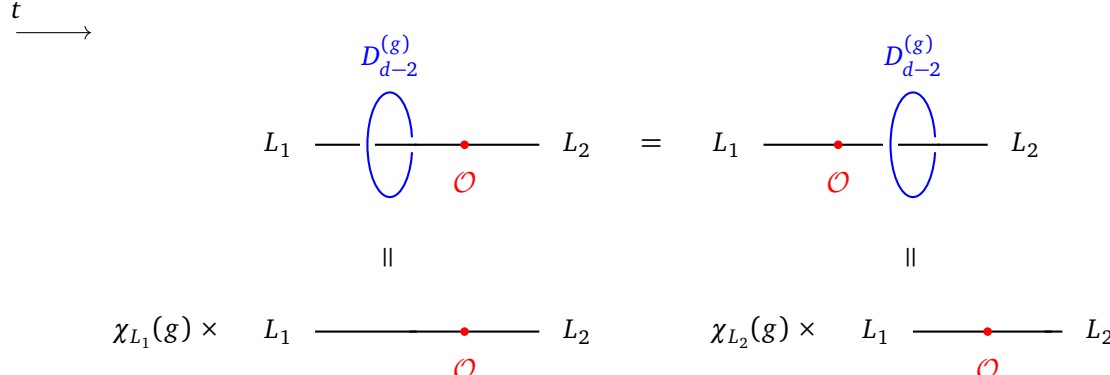

Figure 15: Charge conservation of 1-form symmetry (proof of the absence of screening). Time in the above figures is flowing from left to right as indicated. Since all four diagrams are equal for all $g \in G^{(1)}$, the 1-charges $\chi_{L_1}$ and $\chi_{L_2}$ of $L_1$ and $L_2$ respectively under $G^{(1)}$ 1-form symmetry must be equal. This equality can be interpreted as conservation of the 1-charge under time evolutions.

characters of the 1-form symmetry group $G^{(1)}$.

Mathematically, this is because there do not exist 1-morphisms between two simple objects in the 2-category formed by 2-representations of the 2-group $\mathbb{G}^{(2)}_{G^{(1)}}$ associated to a 1-form symmetry group $G^{(1)}$. Physically, this is the statement of charge conservation for 1-form symmetry as explained in figure 15. This explains the $p = q = r = 1$ piece of statement 1.5.

This fact is usually presented by saying that $L_1$ cannot be screened to another line operator $L_2$, if $L_1$ and $L_2$ have different charges under the 1-form symmetry. In particular, a line operator $L$ carrying a non-trivial charge under 1-form symmetry cannot be completely screened, i.e. cannot be screened to the identity line operator.

## 5 Twisted generalized charges

In this section we study higher-charges formed by operators living in twisted sectors of invertible symmetries. These, as defined in section 1.4, arise at the end of symmetry generators or condensation defects. We will see that the structure of twisted charges is sensitive to the 't Hooft anomalies of the symmetry, even for operators of codimension-2 and higher, which is unlike the case of untwisted charges.

### 5.1 Non-anomalous 0-form symmetries

In this subsection, we begin by studying twisted higher-charges that can arise at the ends of symmetry generators of a $G^{(0)}$ 0-form symmetry group. 0-form symmetries are generated by topological codimension-1 operators, and twisted sector operators are (not necessarily topological) codimension-2 operators living at the ends of these topological codimension-1 operators.

**Two dimensions.** In two spacetime dimensions, the twisted sector operators are local operators. Consider a $g$-twisted sector local operator $\mathcal{O}$ living at the end of a topological line operator $D_1^{(g)}$, with $g \in G^{(0)}$:

$$\underline{D_1^{(g)} \qquad \bullet\, \mathcal{O}} \,, \tag{118}$$

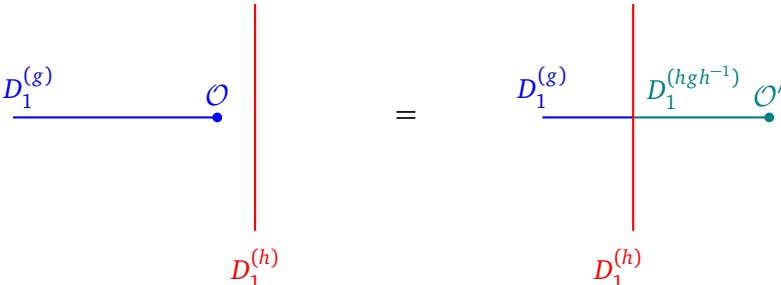

Figure 16: Action of $h \in G^{(0)}$ not in the stabilizer group of $g$, maps a $g$-twisted operator to an $hgh^{-1}$-twisted operator.

where $D_1^{(g)}$ is topological, but $\mathcal{O}$ is generically not. Let

$$H_g = \left\{ h \in G^{(0)} | hgh^{-1} = g \right\}, \tag{119}$$

be the stabilizer subgroup of $g$. We can act on $\mathcal{O}$ by an element

$$h \in G^{(0)}, \qquad h \notin H_g. \tag{120}$$

As explained in figure 16, this maps $\mathcal{O}$ to a local operator $\mathcal{O}'$ living in twisted sector for

$$hgh^{-1} \in [g] \subset G^{(0)}, \tag{121}$$

where $[g]$ is the conjugacy class that $g$ lies in. Consequently, the $g$-twisted sector operator $\mathcal{O}$ lives in a whole (irreducible) multiplet $M$ of operators living in twisted sectors for all elements in the conjugacy class $[g]$. On the other hand, an element

$$h \in H_g \subseteq G^{(0)}, \tag{122}$$

maps $\mathcal{O}$ to a local operator in the $g$-twisted sector, which may be equal to $\mathcal{O}$ or may be some other operator in the $g$-twisted sector.

Putting it all together, we can describe the irreducible multiplet $M$ as a vector space of local operators graded by elements of $[g]$

$$M = \bigoplus_{g \in [g]} M_g, \tag{123}$$

where $M_g$ is the vector space formed by local operators participating in the multiplet $M$ and lying in the $g$-twisted sector. Moreover, the stabilizer $H_g$ acts as linear maps from $M_g$ to itself, implying

$$M_g = \{\text{Irreducible representation of } H_g\}. \tag{124}$$

Similarly, $M_{g'}$ for any $g' \in [g]$ forms an irreducible representation of the corresponding stabilizer group $H_{g'} \subseteq G^{(0)}$. This representation is obtained simply by transporting the representation of $H_g$ formed by $M_g$ using an isomorphism

$$H_g \to H_{g'} = hH_gh^{-1}, \tag{125}$$

induced by

$$g' = hgh^{-1}. \tag{126}$$

This recovers the $d = 2, [\omega_g] = 0$ piece of the statement 1.6.

**Three dimensions.** In three spacetime dimensions, the twisted sector operators for a 0-form symmetry are line operators:

$$D_2^{(g)} \qquad L \; . \tag{127}$$

Line operators $L$ in the $g$-twisted sector for a 0-form symmetry $G^{(0)}$ are part of a multiplet $M$ comprised of line operators, which lie in twisted sectors for elements in the conjugacy class $[g] \subseteq G^{(0)}$.

The stabilizer $H_g$ transforms a $g$-twisted sector line in $M$ back into a $g$-twisted sector line in $M$, but the identity of the $g$-twisted sector line may get transformed as we already discussed for untwisted sector line operators. In total, repeating the arguments of section 2.2 we learn that the line operators in $M$ lying in the $g$-twisted sector form a 2-representation $\rho_g^{(2)}$ of $H_g$, recovering the $d = 3, [\omega_g] = 0$ piece of the statement 1.6.

The line operators in $M$ lying in $g'$-twisted sector for $g' = g'' g g''^{-1}$ form a 2-representation $\rho_{g'}^{(2)}$ of the isomorphic group

$$H_{g'} = g'' H_g g''^{-1} \,, \tag{128}$$

where $\rho_{g'}^{(2)}$ is obtained by transporting $\rho_g^{(2)}$ using the isomorphism provided by the above equation.

**Higher dimensions.** It is now straightforward to see the generalization to an arbitrary spacetime dimension $d$. The twisted sector operators for $G^{(0)}$ symmetry generators have codimension-2 and furnish twisted $(d-2)$-charges. Such an operator $\mathcal{O}_{d-2}$ in $g$-twisted sector forms an irreducible multiplet $M$ with codimension-2 operators lying in twisted sectors for elements in the conjugacy class $[g] \subseteq G^{(0)}$. The stabilizer $H_g$ transforms $g$-twisted sector operators in $M$ back to themselves. In total, the operators in $M$ form a $(d-1)$-representation $\rho_g^{(d-1)}$ of $H_g$, justifying the $[\omega_g] = 0$ piece of the statement 1.6 at the level of the top-most layer of objects. Operators in $M$ lying in another sector in the same conjugacy class $[g]$ are in equivalent $(d-1)$-representations of isomorphic stabilizer subgroups of $G^{(0)}$.

It is also clear that this characterization extends to $(d-3)$-charges going between twisted $(d-2)$-charges, which should form 1-morphisms in the $(d-1)$-category $(d-1)$-Rep$(H_g)$. Similarly, considering lower-dimensional operators, one recovers the full categorical statement made for $[\omega_g] = 0$ in statement 1.6.

### 5.2 Anomalous 0-form symmetry

Let us now turn on a 't Hooft anomaly of the form

$$[\omega] \in H^{d+1}(G^{(0)}, \mathbb{C}^\times), \tag{129}$$

for the bulk $G^{(0)}$ 0-form symmetry and revisit the analysis of the previous subsection.

**Two dimensions.** Just as for the non-anomalous case, the twisted sector operators form multiplets parametrized by conjugacy classes $[g] \in G^{(0)}$, and $g$-twisted sector operators in a $[g]$-multiplet are acted upon by the stabilizer $H_g$. However, instead of forming linear representations of $H_g$, the $g$-twisted operators now form $[\omega_g]$-twisted representations of $H_g$, where

$$\omega_g(h_1, h_2) = \frac{\omega(g, h_1, h_2)\omega(h_1, h_2, g)}{\omega(h_1, g, h_2)} \,, \tag{130}$$

Figure 17: The above chain of equalities provides a formula for the twist $\omega_g = \omega(h_1^{-1}, gh_1, h_2)\omega(g, h_1, h_2)\omega(g^{-1}, h_2^{-1}, h_1^{-1})$ which the reader can verify matches the expression shown in (130). The various $\omega$ factors arise by performing associativity/F-moves on the topological line operators generating an anomalous 0-form symmetry in 2d.

for all $h_1, h_2 \in H_g$. See figure 17 for explanation and section 4.1 for more details on twisted representations. The map

$$H^3(G^{(0)}, \mathbb{C}^\times) \to H^2(H_g, \mathbb{C}^\times), \tag{131}$$

induced by

$$[\omega] \mapsto [\omega_g], \tag{132}$$

is often referred to as *slant product* in the literature (see e.g. [69]).

This justifies $d = 2$ piece of the statement 1.6.

**Three dimensions.** Again, as in the non-anomalous case, the line operators in $g$-twisted sector form multiplets $M$. The stabilizer $H_g$ still sends $g$-twisted sector lines into each other. The associativity of the action of $H_g$ is governed by

$$[\omega_g] \in H^3(H_g, \mathbb{C}^\times), \tag{133}$$

which is obtained from $[\omega]$ via

$$\omega_g(h_1, h_2, h_3) = \frac{\omega(g, h_1, h_2, h_3)\omega(h_1, h_2, g, h_3)}{\omega(h_1, g, h_2, h_3)\omega(h_1, h_2, h_3, g)}. \tag{134}$$

The class $[\omega_g]$ is again referred to as the slant product of $[\omega]$ and $g$.

Now, consider a line operator $L$ in the $g$-twisted sector and let $H_L \subseteq H_g$ be the group that sends $L$ to itself. Physically, $H_L$ must descend to an induced 0-form symmetry group of line $L$, which imposes constraints on the allowed possibilities for $H_L$ as a subgroup of $H_g$. To see these constraints, pick an arbitrary topological local operator $D_0^{(h)}$ in the junction of $L$ with $D_{d-1}^{(h)}$ for all $h \in H_L$. As we fuse these operators, we will in general generate factors $\sigma(h_1, h_2) \in \mathbb{C}^\times$

$$D_0^{(h_1)} \otimes D_0^{(h_2)} = \sigma(h_1, h_2) D_0^{(h_1 h_2)}. \tag{135}$$

The action of the induced $H_L$ 0-form symmetry must be associative, which means that the non-associativity factor arising from $\omega_g$ must be cancelled by the $\sigma$ factors as follows

$$\omega_g(h_1, h_2, h_3) = \frac{\sigma(h_2, h_3)\sigma(h_1, h_2 h_3)}{\sigma(h_1 h_2, h_3)\sigma(h_1, h_2)} = \delta\sigma(h_1, h_2, h_3), \tag{136}$$

where $h_1, h_2, h_3 \in H_L$. See figure 18. In particular $H_L$ must be such that

$$[\omega_g]_{H_L} = 0 \in H^3(H_L, \mathbb{C}^\times), \tag{137}$$

where $[\omega_g]_{H_L}$ is the restriction of $[\omega_g]$ to $H_L$. That is, $\sigma$ is a *trivialization* of $\omega_g$.

There is additional information in the factors $\sigma$. Note that by redefining topological local operators $D_0^{(h)}$, we can redefine $\sigma$ as

$$\sigma \to \sigma' = \sigma + \delta\alpha, \tag{138}$$

for a $\mathbb{C}^\times$ valued 1-cochain $\alpha$ on $H_L$. This means that two 1-charges differentiated only by having 2-cochains $\sigma$ and $\sigma'$ such that

$$\sigma' \neq \sigma + \delta\alpha, \tag{139}$$

for all 1-cochains $\alpha$ describe different 1-charges. Physically, the equivalence class $[\sigma]$ of $\mathbb{C}^\times$-valued 2-cochains on $H_L$ under the equivalence relation (138) and satisfying (136) should be viewed as capturing a 't Hooft anomaly for the $H_L$ 1-form symmetry.

In fact, the information of:

1. $H_L \subseteq H_g$ satisfying (137), and

2. a choice of equivalence class $[\sigma]$ with any representative $\sigma$ satisfying (136),

specifies a $[\omega_g]$-twisted 2-representation of $H_g$. Thus, we have justified $d = 3$ piece of statement 1.6 at the level of objects.

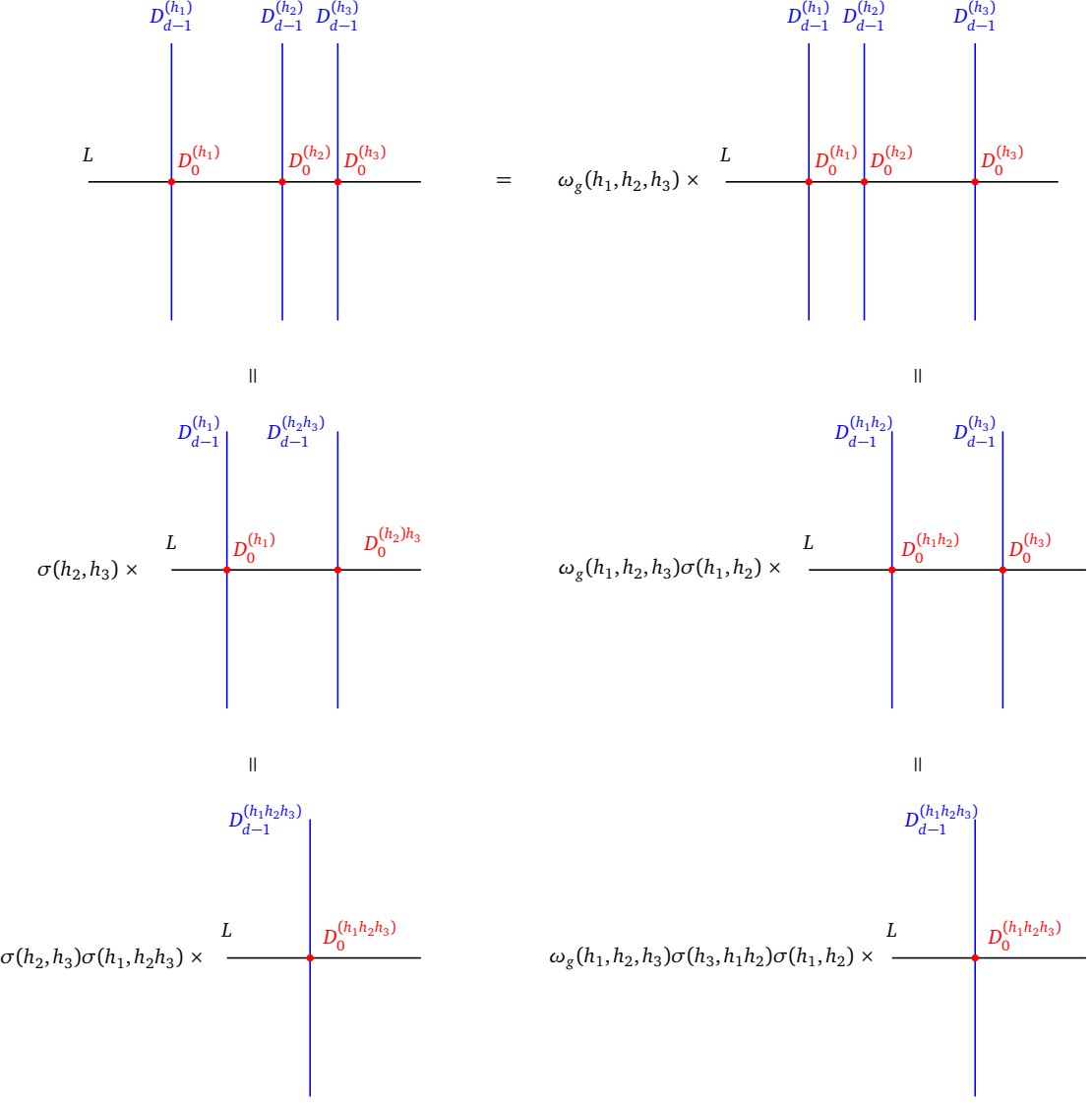

Figure 18: Depiction of equation (136), which results from following the chain of equalities equating bottom-left and bottom-right parts of the above figure. Even though the line operator $L$ is in the twisted sector, for brevity we have dropped the topological surface operator that it is attached to.

**Higher dimensions.** To justify statement 1.6 in full generality (i.e. for higher $d$ and also higher categorical levels), we can adopt the following approach. The topological operator $D_{d-1}^{(g)}$ must form a $(d-1)$-charge of $H_g$. Then the twisted higher-charges that can arise at the end of $D_{d-1}^{(g)}$ must form the $(d-1)$-category formed by morphisms from this $(d-1)$-charge to the trivial $(d-1)$-charge inside the $d$-category formed by possible higher-charges of codimension-1 and higher-codimensional operators under $H_g$ 0-form symmetry.

We claim that the $(d-1)$-charge under $H_g$ formed by $D_{d-1}^{(g)}$ is specified by a $d$-representation $\rho_{[\omega_g]}^{(d)}$ of $H_g$ that does not exhibit symmetry fractionalization. Recall from section 2.3.1 that such $d$-representations are characterized by the choice of a subgroup $H \subseteq H_g$ and a class $[\sigma] \in H^d(H, \mathbb{C}^\times)$. The $d$-representation $\rho_{[\omega_g]}^{(d)}$ is specified by

$$H = H_g, \qquad [\sigma] = [\omega_g], \tag{140}$$

and is actually a 1-dimensional $d$-representation of group cohomology type. Here

$$[\omega_g] \in H^d(H_g, \mathbb{C}^\times), \tag{141}$$

is obtained by performing a slant product of $g$ with $[\omega]$

$$\omega_g(h_1, h_2, \cdots, h_d) = \prod_{i=0}^{d} \omega^{s(i)}(h_1, \cdots, h_i, g, h_{i+1}, \cdots, h_d), \tag{142}$$

where $h_i \in H_g$, $s(i) = 1$ for even $i$ and $s(i) = -1$ for odd $i$.

Thus twisted $g$-sector generalized charges are specified by the $(d-1)$-category of morphisms from $\rho_{[\omega_g]}^{(d)}$ to identity $d$-representation in the $d$-category $d$-$\mathsf{Rep}(H_g)$ formed by $d$-representations of $H_g$. We denoted this $(d-1)$-category as

$$(d-1)\text{-}\mathsf{Rep}^{[\omega_g]}(H_g), \tag{143}$$

in statement 1.6 and called its objects as '$[\omega_g]$-twisted $(d-1)$-representations of $H_g$'. This is because for low $d$, this matches the more well-known notion of twisted representations and twisted 2-representations, which has been discussed in detail above.

## 5.3 Mixed 't Hooft anomaly between 1-form and 0-form symmetries

We have seen above that in the presence of 't Hooft anomaly, the structure of twisted generalized charges is quite different from the structure of untwisted generalized charges. In this subsection, we will see another example of this phenomenon, while studying the structure of 1-charges in the presence of 1-form and 0-form symmetries with a mixed 't Hooft anomaly in 4d QFTs.

In particular, we consider in 4d, 0-form and 1-form symmetries

$$G^{(0)} = \mathbb{Z}_2^{(0)}, \qquad G^{(1)} = \mathbb{Z}_2^{(1)}, \tag{144}$$

with mixed 't Hooft anomaly

$$\mathcal{A}_5 = \exp\left(\pi i \int A_1 \cup \frac{\mathcal{P}(B_2)}{2}\right). \tag{145}$$

Let $D_2$ and $D_3$ be the topological operators generating $\mathbb{Z}_2^{(1)}$ and $\mathbb{Z}_2^{(0)}$ respectively. Let us encode the anomaly (145) in terms of properties of these operators. This will help us later in

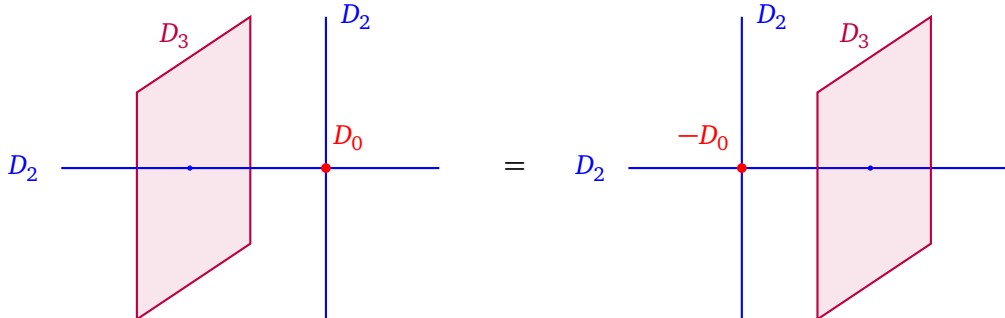

Figure 19: The mixed 0-form/1-form symmetry anomaly (145) as seen from the topological defects. The junction $D_0$ between the two topological surface defects $D_2$ that generate the 1-form symmetry is charged under the 0-form symmetry generated by the codimension-1 topological defect $D_3$.

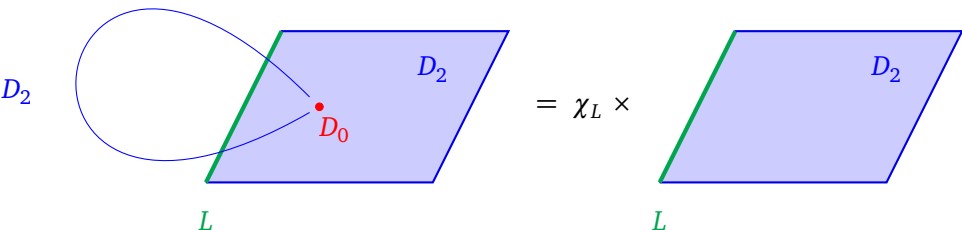

Figure 20: Character assigned to the twisted sector line operator $L$ depends on a choice of topological local operator $D_0$ satisfying the property $(D_0)^2 = 1$.

exploring the impact of the anomaly on the structure of twisted 1-charges for $\mathbb{Z}_2^{(1)}$. A transversal intersection of two $D_2$ operators leads to a point-like junction housing a 1-dimensional space of topological local operators. The anomaly (145) states that these junction local operators are non-trivially charged under $\mathbb{Z}_2^{(0)}$. That is, as we move $D_3$ across such a topological junction local operator $D_0$, we implement the action

$$D_0 \to -D_0 \,. \tag{146}$$

See figure 19.

Now consider a line operator $L$ that can arise at the end of $D_2$. Pick a topological junction local operator $D_0$ satisfying

$$(D_0)^2 = 1 \,, \tag{147}$$

so that $D_0$ can be viewed as implementing a $\mathbb{Z}_2$ induced 1-form symmetry on $D_2$. Note that there are two possible choices of $D_0$ differing by a sign, and we have made one of the two choices. Once such a choice of $D_0$ has been made, we can assign a character of $\mathbb{Z}_2^{(1)}$ to the twisted sector line operator $L$ as shown in figure 20, which may be interpreted as the charge of $L$ under the $\mathbb{Z}_2^{(1)}$ 1-form symmetry. Note though that being charged or not is not a fundamental property of $L$, but rather dependent on the choice of $D_0$:

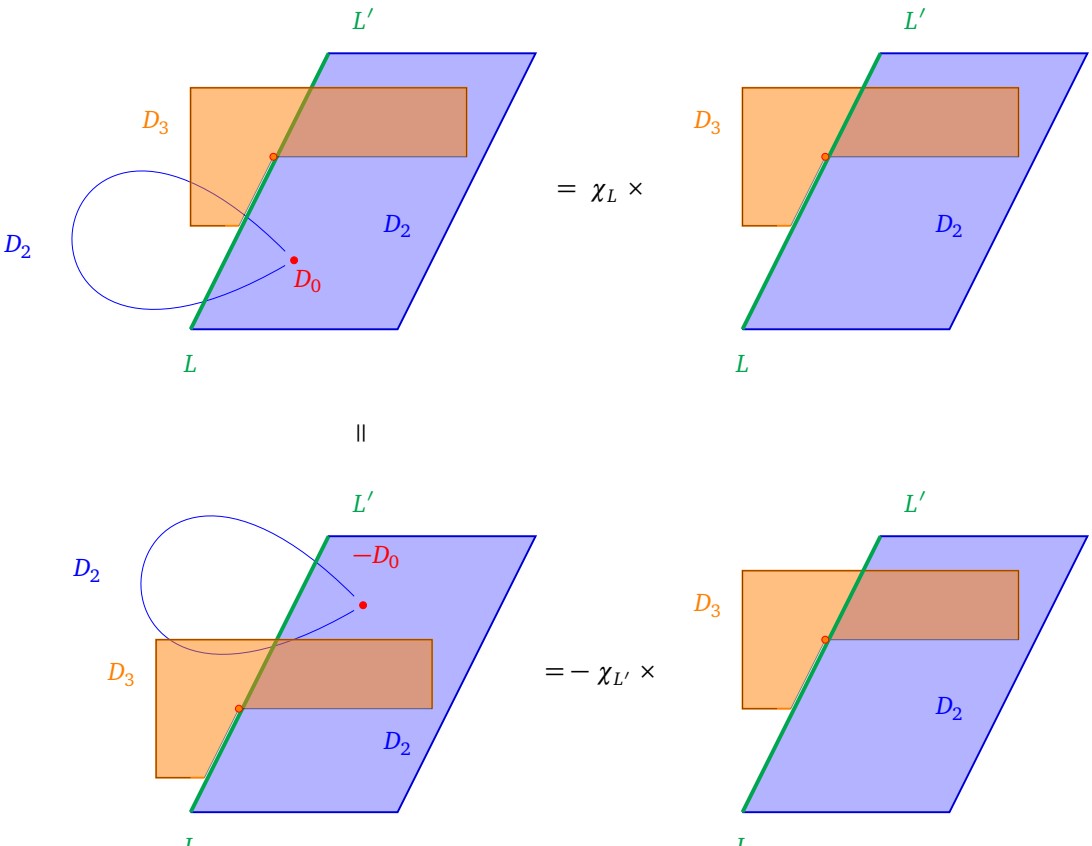

Figure 21: In the above figure, we study the possible action of a $\mathbb{Z}_2^{(0)}$ 0-form symmetry generated by $D_3$ on line operators living at the end of the topological surface operator $D_2$ generating $\mathbb{Z}_2^{(1)}$ 1-form symmetry. Let $D_3$ act on line operator $L$ and transform into a line operator $L'$, which may or may not be the same as $L$. Now we pick a choice of topological junction local operator $D_0$ and compare the characters carried by $L$ and $L'$. The chain equalities shown above forces us to conclude that $\chi_L = -\chi_{L'}$, that is $L$ and $L'$ must have opposite charges under 1-form symmetry, and in particular $L \neq L'$.

1. If $L$ is charged when the choice $D_0$ is made, then $L$ is uncharged when the choice $-D_0$ is made.

2. If $L$ is uncharged when the choice $D_0$ is made, then $L$ is charged when the choice $-D_0$ is made.

In any case, according to (146) the choice of $D_0$ is modified when passing it through $D_3$, and so the type of line operator $L$ cannot be left invariant when it passes through $D_3$. See figure 21 for more details. If $L$ is charged (uncharged) under a fixed choice of $D_0$, then it must be modified to a line operator $L'$ uncharged (charged) under the same choice of $D_0$.

Thus a there is a single possibility for an irreducible 1-charge lying in twisted sector of $\mathbb{Z}_2^{(1)}$. This 1-charge comprises of a multiplet of two simple line operators $L$ and $L'$, such that the action of $\mathbb{Z}_2^{(1)}$ on $L$ and $L'$ differs by a sign. Below we study two examples of 4d QFTs having this symmetry and anomaly structure, where only such twisted 1-charges appear. This not only is consistent with the above general result but also explains why the same physical phenomenon is observed in both theories.

**Example 5.1: 4d $\mathcal{N} = 1$ super-Yang–Mills**

These symmetries and anomalies are realized in 4d $\mathcal{N} = 1$ $SU(2)$ SYM, where $\mathbb{Z}_2^{(1)}$ is the electric/center 1-form symmetry and $\mathbb{Z}_2^{(0)}$ quotient of the $\mathbb{Z}_4^{(0)}$ chiral/R-symmetry (only the $\mathbb{Z}_2^{(0)}$ has a mixed anomaly).

We have three unscreened line operators to consider: denoted typically as $W$, $H$ and $W + H$. The line operator $W$ can be realized as Wilson line transforming in fundamental representation of $SU(2)$ gauge group. The line operator $H$ can be realized as 't Hooft line that induces an $SO(3)$ monopole configuration on a sphere $S^2$ linking it, which cannot be lifted to an $SU(2)$ monopole configuration. The line operator $W + H$ can be realized as a dyonic line that carries both fundamental representation and induces an $SO(3)$ monopole configuration, and can be understood as being obtained by combining the $W$ and $H$ lines.

The line $W$ lies in the untwisted sector and is charged non-trivially under $\mathbb{Z}_2^{(1)}$. On the other hand, the lines $H$ and $W + H$ both lie in the twisted sector, arising at the end of topological operator $D_2$. The charges of $H$ and $W + H$ under $\mathbb{Z}_2^{(1)}$ must be opposite because they are related by the line $W$ which is charged under $\mathbb{Z}_2^{(1)}$.

Finally, note that $H$ and $W + H$ are indeed permuted into each other by the generator of the $\mathbb{Z}_4^{(0)}$ chiral symmetry, as it is well-known that it implements the Witten effect.

**Example 5.2: 4d $\mathcal{N} = 4$ super-Yang-Mills**

These symmetries and anomalies are also realized in 4d $\mathcal{N} = 4$ $SO(3)_-$ SYM at the self-dual point ($\tau = -1/\tau$), where $\mathbb{Z}_2^{(1)}$ is the dyonic 1-form symmetry and $\mathbb{Z}_2^{(0)}$ is the S-duality that acts as a symmetry at the self-dual point.

We again have the same three unscreened line operators to consider: $W$, $H$ and $W + H$. The line $W + H$ lies in the untwisted sector and is charged non-trivially under $\mathbb{Z}_2^{(1)}$. On the other hand, the lines $H$ and $W$ both lie in the twisted sector, arising at the end of topological operator $D_2$. The charges of $H$ and $W$ under $\mathbb{Z}_2^{(1)}$ must be opposite because they are related by the line $W + H$ which is charged under $\mathbb{Z}_2^{(1)}$.

Finally, note that $H$ and $W$ are indeed permuted into each other by the $\mathbb{Z}_2^{(0)}$ S-duality.

It is straightforward to generalize to general $G^{(0)}$ and $G^{(1)}$, but the expression for the anomaly (145) takes a more complicated form involving a cohomological operation combining the cup product and Pontryagin square operations appearing in (145) into a single operation, which takes in $A_1$ and $B_2$ to output the anomaly $\mathcal{A}_5$ directly.

We will see in Part II [1] that this fact leads to a well-known action [15, 16, 29] of a non-invertible symmetry on line operators, permuting untwisted sector and twisted sector lines for a 1-form symmetry into each other.

## 5.4 Condensation twisted charges

In this subsection, we study generalized charges appearing in twisted sectors associated to condensation defects. As described in the definition in section 1.4, condensation defects are topological defects obtained by gauging invertible symmetry generating topological defects on submanifolds in spacetime.

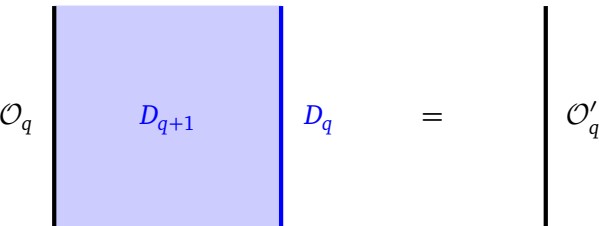

Figure 22: $D_{q+1}$ is a $(q + 1)$-dimensional condensation defect, which means that it admits at least one topological non-genuine $q$-dimensional defect living at its boundary. In the figure, $D_q$ is one such topological boundary of $D_{q+1}$. On the other hand $\mathcal{O}_q$ is a possibly non-topological operator living on the boundary of $D_{q+1}$. In other words, $\mathcal{O}_q$ is in the twisted sector for the condensation defect $D_{q+1}$. We can perform an interval compactification involving $\mathcal{O}_q$, $D_{q+1}$ and $D_q$ as shown in the figure to obtain an untwisted sector possibly non-topological $q$-dimensional operator $\mathcal{O}'_q$.

**Twisted to untwisted.** The first interesting physical observation here is that a $q$-dimensional operator in twisted sector for a $(q + 1)$-dimensional condensation defect can always be converted into a $q$-dimensional untwisted sector operator. This is because a condensation defect always admits a topological end, which allows us to perform the above transition as explained in figure 22. There might be multiple such topological ends and hence multiple ways of performing the above transition. However, one should note that there is a canonical topological end as well corresponding to Dirichlet boundary conditions for the gauge fields localized on the $(q + 1)$-dimensional locus occupied by the condensation defect. Below we assume that we have performed this transition using this canonical boundary condition.

**Untwisted to twisted.** One may now ask if all untwisted sector operators can be obtained this way. This is not true, as we illustrate through the case of a non-anomalous $p$-form symmetry $G^{(p)}$ with $p \geq 1$ in the bulk. In this situation, only $q$-dimensional untwisted sector operators, on which the bulk $p$-form symmetry descends to an induced $r$-form symmetry for some $r < p$, can be obtained from $q$-dimensional twisted sector operators for condensation defects.

More precisely consider an untwisted sector $q$-dimensional operator $\mathcal{O}_q$ for

$$q = d + r - p - 1, \tag{148}$$

on which a subgroup

$$G_{\mathcal{O}_q}^{(r)} \subseteq G^{(p)}, \tag{149}$$

descends to an induced $r$-form symmetry. In particular, the $q$-dimensional operator $\mathcal{O}_q$ is invariant under the subgroup $G_{\mathcal{O}_q}^{(r)}$.

The induced $r$-form symmetry may have a 't Hooft anomaly which we can express as follows. First of all, pick a background field $B_{r+1}|_{\mathcal{O}_q}$ for the induced $G_{\mathcal{O}_q}^{(r)}$ symmetry living on the world-volume of $\mathcal{O}_q$. Since topological operators generating induced symmetries extend into the bulk, we need to also specify a background for the $G^{(p)}$ bulk $p$-form symmetry. There is a canonical way of specifying such a background by restricting the bulk topological operators generating $G^{(p)}$ to only lie in a $(q + 1)$-dimensional submanifold $\Sigma_{q+1}$ of $d$-dimensional spacetime, whose boundary is the world-volume of $\mathcal{O}_q$. This gives rise to an $r$-form symmetry background $B_{r+1}|_{\Sigma_{q+1}}$ on $\Sigma_{q+1}$ whose restriction to the world-volume of $\mathcal{O}_q$ gives rise to the background $B_{r+1}|_{\mathcal{O}_q}$.

Now, performing gauge transformations of the background $B_{r+1}|_{\Sigma_{q+1}}$, we might find a 't Hooft anomaly for the induced $G_{\mathcal{O}_q}^{(r)}$ symmetry taking the form

$$\mathcal{A}_{q+1} = \exp\left(2\pi i \int_{\Sigma_{q+1}} \Theta_{q+1}\left(B_{r+1}|_{\Sigma_{q+1}}\right)\right), \tag{150}$$

where $\Theta_{q+1}\left(B_{r+1}|_{\mathcal{O}_q}\right)$ is an $\mathbb{R}/\mathbb{Z}$-valued cochain on $\Sigma_{q+1}$, which is a function of the background field $B_{r+1}|_{\Sigma_{q+1}}$. This can be canceled by adding along $\Sigma_{q+1}$ a $G_{\mathcal{O}_q}^{(r)}$ protected SPT phase whose effective action is $\mathcal{A}_{q+1}$.

Once the anomaly has been cancelled in this fashion, we can promote $B_{r+1}|_{\Sigma_{q+1}}$ to a dynamical gauge field that we denote as $b_{r+1}|_{\Sigma_{q+1}}$, thus gauging the $G_{\mathcal{O}_q}^{(r)}$ symmetry on $\Sigma_{q+1}$ and the world-volume of $\mathcal{O}_q$. The additional SPT phase (150) becomes a Dijkgraaf-Witten twist or discrete torsion for the gauging. After gauging, we obtain a condensation defect

$$D_{q+1}^{(G_{\mathcal{O}_q}^{(r)}, \Theta_{q+1})}, \tag{151}$$

placed along $\Sigma_{q+1}$ and the untwisted sector operator $\mathcal{O}_q$ has been promoted to an operator $\mathcal{O}_q'$ lying in the twisted sector for the condensation defect (151).

Finally, we can transition from twisted sector $\mathcal{O}_q'$ back to untwisted sector $\mathcal{O}_q$ by placing a Dirichlet boundary condition at another end of (151) as already explained in figure 22.

**Simplest case: $p = 1$**  The above general analysis can be neatly illustrated for a $G^{(1)}$ 1-form symmetry in the bulk generated by topological codimension-2 operators $D_{d-2}^{(g)}$ for $g \in G^{(1)}$. The only possibility is $r = 0$ corresponding to $q = d-2$. Consider an untwisted sector codimension-2 operator $\mathcal{O}_{d-2}$. It has an induced symmetry

$$G_{\mathcal{O}_{d-2}}^{(0)} \subseteq G^{(1)}, \tag{152}$$

if we have

$$D_{d-2}^{(g)} \otimes \mathcal{O}_{d-2} = \mathcal{O}_{d-2}, \qquad \forall\, g \in G_{\mathcal{O}_{d-2}}^{(0)} \subseteq G^{(1)}, \tag{153}$$

i.e. if $\mathcal{O}_{d-2}$ is left invariant by topological codimension-2 operators generating $G_{\mathcal{O}_{d-2}}^{(0)}$ subgroup of 1-form symmetry group $G^{(1)}$.

The $G_{\mathcal{O}_{d-2}}^{(0)}$ induced symmetry may have a 't Hooft anomaly described by

$$[\Theta] \in H^{d-1}(G_{\mathcal{O}_{d-2}}^{(0)}, \mathbb{C}^{\times}). \tag{154}$$

In this case $\mathcal{O}_{d-2}$ can be promoted to $\mathcal{O}_{d-2}'$, which is a codimension-2 operator living in the twisted sector for the condensation defect obtained by gauging the $G_{\mathcal{O}_{d-2}}^{(0)}$ subgroup of $G^{(1)}$ bulk 1-form symmetry along a codimension-1 manifold with Dijkgraaf-Witten twist $[\Theta]$.

---

**Example 5.3: Pure Pin$^+$(4N) gauge theory**

Begin with pure Spin($4N$) gauge theory in any spacetime dimension $d$, which has

$$G^{(1)} = \mathbb{Z}_2^{(S)} \times \mathbb{Z}_2^{(C)}, \tag{155}$$

center 1-form symmetry. Let us denote the topological operators generating $\mathbb{Z}_2^{(S)}$ and $\mathbb{Z}_2^{(C)}$ as $D_{d-2}^{(S)}$ and $D_{d-2}^{(C)}$ respectively and the topological operator generating the diagonal $\mathbb{Z}_2$

in $G^{(1)}$ as $D_{d-2}^{(V)}$. There is additionally a 0-form outer automorphism symmetry

$$G^{(0)} = \mathbb{Z}_2^{(0)}, \tag{156}$$

that exchanges $\mathbb{Z}_2^{(S)}$ and $\mathbb{Z}_2^{(C)}$.

Let us gauge $\mathbb{Z}_2^{(0)}$, which takes us to the $d$-dimensional pure Pin$^+(4N)$ gauge theory theory, which has a simple non-invertible codimension-2 topological operator $D_{d-2}^{(SC)}$ arising as the image of the non-simple codimension-2 topological operator

$$D_{d-2}^{(S)} \oplus D_{d-2}^{(C)}, \tag{157}$$

of the Spin$(4N)$ theory. Note that $D_{d-2}^{(S)}$ and $D_{d-2}^{(C)}$ do not descend to topological operators of the Pin$^+(4N)$ theory, because they are not left invariant by the action of $\mathbb{Z}_2^{(0)}$; however, (157) is left invariant by $\mathbb{Z}_2^{(0)}$ and descends to the topological operator $D_{d-2}^{(SC)}$ of the Pin$^+(4N)$ theory. Additionally, $D_{d-2}^{(V)}$ also descends to a topological operator of the Pin$^+(4N)$ theory because it is not acted upon by $\mathbb{Z}_2^{(0)}$. In the Pin$^+(4N)$ theory, we have the fusion rule

$$D_{d-2}^{(V)} \otimes D_{d-2}^{(SC)} = D_{d-2}^{(SC)}, \tag{158}$$

satisfying the condition (153) with $\mathcal{O}_{d-2} = D_{d-2}^{(SC)}$ and $G_{\mathcal{O}_{d-2}}^{(0)} = G^{(1)} = \mathbb{Z}_2^{(V)}$ generated by $D_{d-2}^{(V)}$.

This means we can convert the untwisted sector codimension-2 topological operator $D_{d-2}^{(SC)}$ to a codimension-2 topological operator $D_{d-2}^{(SC;\mathbb{Z}_2)}$ living at the end of the condensation defect

$$D_{d-1}^{(\mathbb{Z}_2)}, \tag{159}$$

obtained by gauging the $\mathbb{Z}_2^{(V)}$ 1-form symmetry of the Pin$^+(4N)$ theory on a codimension-1 manifold in spacetime.

# 6  1-charges for 2-group symmetries

In this section, we study possible 1-charges that can be furnished by line operators under an arbitrary 2-group symmetry. A 2-group symmetry combines 0-form and 1-form symmetries, encapsulating possible interactions between the two types of symmetries.

We will proceed by studying 2-groups of increasing complexity. Let us begin by addressing "trivial" 2-groups, in which there are no interactions between the 0-form and 1-form symmetries. Then a 1-charge of the 2-group is a tuple formed by an arbitrary 1-charge of the $G^{(0)}$ 0-form symmetry and an arbitrary 1-charge of the $G^{(1)}$ 1-form symmetry without any correlation between these two pieces of data.

## 6.1  Split 2-group symmetry

The simplest possible interaction between 0-form symmetry and 1-form symmetry arises when 0-form symmetry acts on 1-form symmetry generators by changing their type. See figure 23. That is we have a collection of automorphisms

$$\alpha_g : G^{(1)} \to G^{(1)}, \tag{160}$$

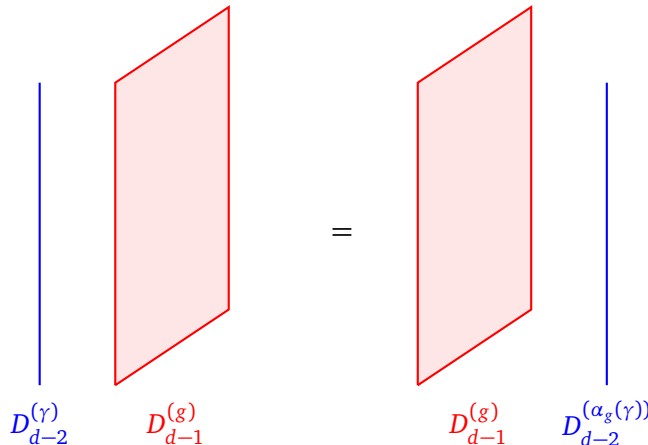

**Figure 23:** Action of the 0-form symmetry $D_{d-1}^{(g)}$ on the 1-form symmetry generated by $D_{d-2}^{(\gamma)}$.

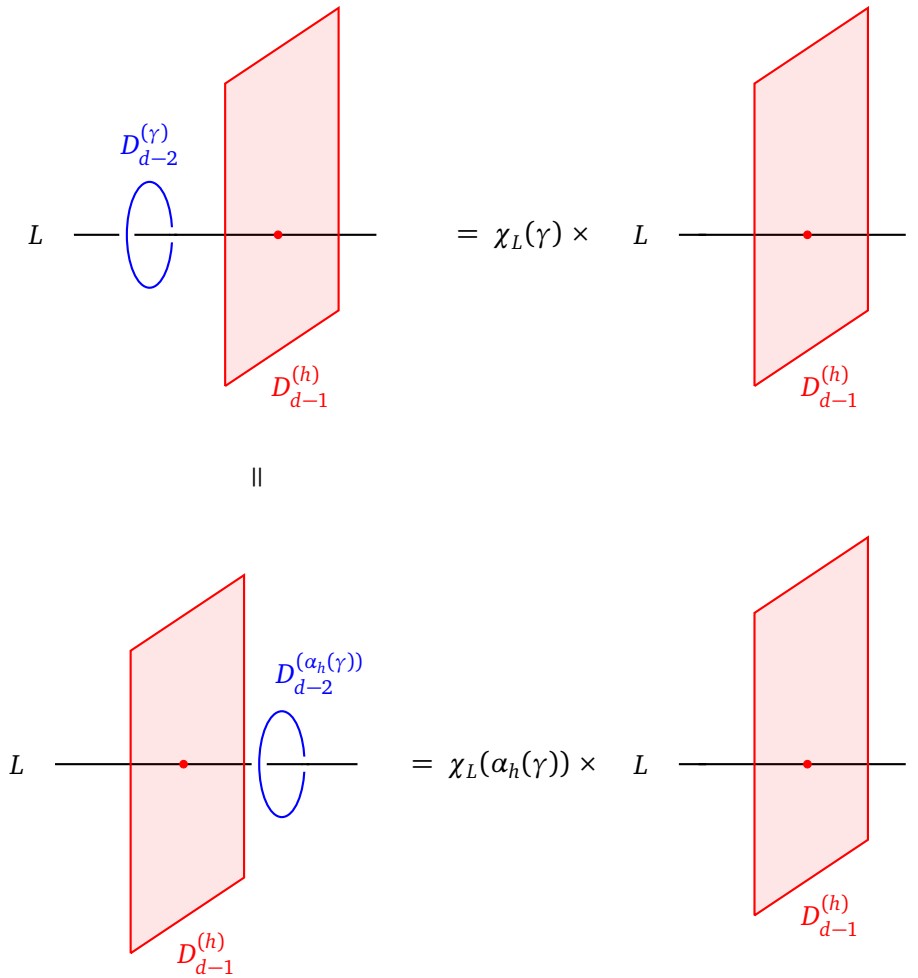

**Figure 24:** The chain of equalities displayed above imposes the condition (162) restricting the possible characters that can be carried by a line operator $L$ under 1-form symmetry $G^{(1)}$ if $H \subseteq G^{(0)}$ bulk 0-form symmetry descends to an induced 0-form on $L$.

of $G^{(1)}$ labeled by elements $g \in G^{(0)}$ such that

$$\alpha_g \alpha_{g'} = \alpha_{gg'}. \tag{161}$$

If this is the only interaction between 0-form and 1-form symmetries, then such a 2-group is known as a *split* 2-group.

Now let us study the action of such a split 2-group on line operators. Acting just by the 0-form symmetry, the line operators must form 2-representations of $G^{(0)}$. Let us pick a line operator $L$ participating in a 2-representation $G^{(0)}$ such that $H_L \subseteq G^{(0)}$ descends to an induced 0-form symmetry of $L$ with a 't Hooft anomaly $[\sigma] \in H^2(H_L, \mathbb{C}^\times)$: $\rho^{(2)} = (H_L, [\sigma])$.

Now, we ask what are the possible characters $\chi_L \in \widehat{G}^{(1)}$ of $G^{(1)}$ that $L$ can carry. Since $H_L$ leaves $L$ invariant, we must have

$$\chi_L(\gamma) = \chi_L(\alpha_h(\gamma)), \tag{162}$$

for all $h \in H_L$ and $\gamma \in G^{(1)}$. See figure 24. Such characters form a subgroup $\widehat{G}^{(1)}_{H_L} \subseteq \widehat{G}^{(1)}$.

An equivalent mathematical characterization of $\widehat{G}^{(1)}_{H_L}$ is as follows. First, note that the action $\alpha$ of $G^{(0)}$ on $G^{(1)}$ induces a dual action $\widehat{\alpha}$ of $G^{(0)}$ on $\widehat{G}^{(1)}$ satisfying

$$\chi(\gamma) = \widehat{\alpha}_g(\chi)(\alpha_g(\gamma)), \tag{163}$$

for all $g \in G^{(0)}$, $\chi \in \widehat{G}^{(1)}$ and $\gamma \in G^{(1)}$. Then $\widehat{G}^{(1)}_{H_L}$ is the subgroup of $\widehat{G}^{(1)}$ formed by elements left invariant by $\widehat{\alpha}_h$ for all $h \in H_L$.

Using the dual action, it is straightforward to describe the character of $G^{(1)}$ carried by another line operator $L_{[g]}$ in the multiplet $M_L$. If $L$ carries character $\chi_L \in \widehat{G}^{(1)}_{H_L}$, then $L_{[g]}$ carries character

$$\widehat{\alpha}_g(\chi_L) \in \widehat{G}^{(1)}_{H_{L_{[g]}}}. \tag{164}$$

Thus the action of the split 2-group on a multiplet $M_L$ of line operators is described by

$$\boldsymbol{\rho}^{(2)} = (H_L, [\sigma], \chi_L), \qquad H_L \subseteq G^{(0)}, \qquad [\sigma] \in H^2(H_L, \mathbb{C}^\times), \qquad \chi_L \in \widehat{G}^{(1)}_{H_L}, \tag{165}$$

which precisely specifies an irreducible 2-representation of the split 2-group, thus justifying a a part of the $p = 2, q = 1$ piece of statement 1.4.

## 6.2 2-group symmetry with untwisted Postnikov class

A different kind of 2-group symmetry arises when there is no action of 0-form symmetry on 1-form symmetry, but the associativity of 0-form symmetry is modified by 1-form symmetry. This modification is captured by an element

$$[\Theta] \in H^3(G^{(0)}, G^{(1)}), \tag{166}$$

which is known as the *Postnikov class* associated to the 2-group symmetry. Since there is no action of $G^{(0)}$ on $G^{(1)}$, the Postnikov class is an element of the untwisted cohomology group.

In understanding the action of such a 2-group on line operators, we follow the same procedure as above. Let $M_L$ be a multiplet/orbit of line operators formed under the permutation action of $G^{(0)}$ starting from a line operator $L$, and let $H_L \subseteq G^{(0)}$ be the 0-form symmetry induced on $L$. Furthermore, let $\chi_L \in \widehat{G}^{(1)}$ be the charge of $L$ under the 1-form symmetry $G^{(1)}$. As we try to check the associativity the $H_L$ action on $L$, we generate a 1-form symmetry transformation on $L$, as discussed in figure 25, leading to an additional factor $\chi_L(\Theta(h_1, h_2, h_3)) \in \mathbb{C}^\times$. For

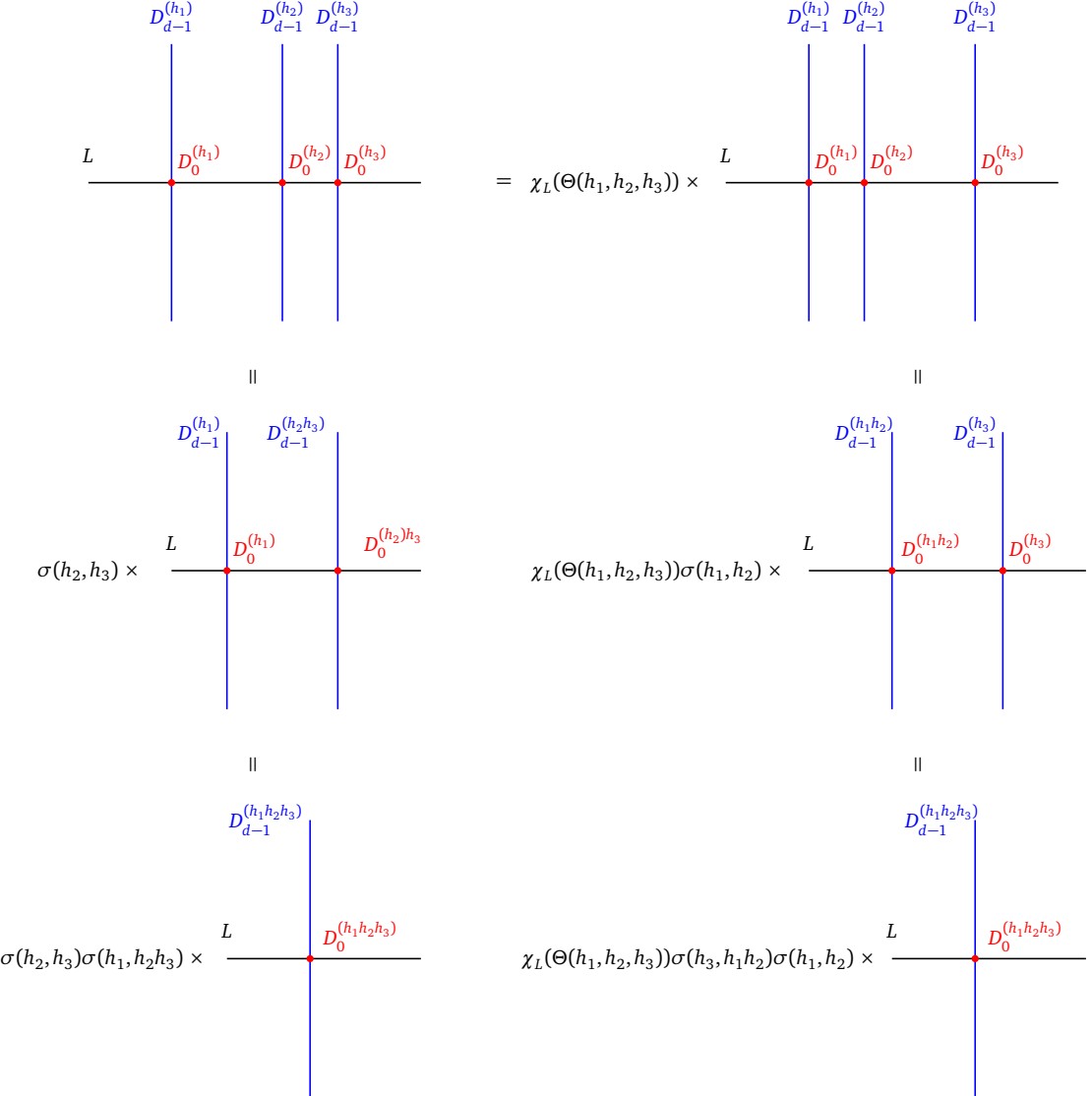

Figure 25: The chain of equalities leading to the condition (167). The equality of top-left and top-right parts of the figure can be taken as the definition of the 2-group symmetry being studied here.

maintaining associativity, we need to cancel these factors by allowing the topological junction local operators to produce extra factors $\sigma(h_1, h_2) \in \mathbb{C}^\times$ that cancel the above factor $\chi_L(\Theta(h_1, h_2, h_3))$. As explained in figure 25, the cancellation condition takes the form

$$\chi_L(\Theta(h_1, h_2, h_3)) = \delta\sigma(h_1, h_2, h_3). \tag{167}$$

This imposes a correlation between the choices of $(H_L, \chi_L)$ which states that

$$\chi_L\big([\Theta]_{H_L}\big) = 0 \in H^3(H_L, U(1)), \tag{168}$$

where $[\Theta]_{H_L}$ is the restriction of $[\Theta]$ from $G^{(0)}$ to $H_L$.

Thus, an irreducible 1-charge of a 2-group symmetry carrying Postnikov class $[\Theta]$ is specified by

1. A subgroup $H_L \subseteq G^{(0)}$ and a character $\chi_L \in \widehat{G}^{(1)}$ satisfying the condition (168).

2. A class $[\sigma]$ of $\mathbb{C}^\times$ valued 2-cochains on $H_L$, such that any 2-cochain $\sigma$ in the class $[\sigma]$ satisfies the equation (167), and two 2-cochains $\sigma, \sigma' \in [\sigma]$ are related by

$$\sigma' = \sigma + \delta\beta \,, \tag{169}$$

where $\beta$ is a $\mathbb{C}^\times$ valued 1-cochain on $H_L$.

This data

$$\rho^{(2)} = (H_L, [\sigma], \chi_L) \,, \tag{170}$$

characterizes precisely the irreducible 2-representations of the 2-group with Postnikov class $[\Theta]$ under discussion [26, 66], justifying a part of the $p = 2, q = 1$ piece of statement 1.4.

## 6.3 General 2-group symmetry

A general 2-group $\mathbb{G}^{(2)}$ has the following information

$$\mathbb{G}^{(2)} = \left(G^{(0)}, G^{(1)}, \alpha, [\Theta]\right) \,, \tag{171}$$

where $G^{(0)}$ is a 0-form symmetry group, $G^{(1)}$ is a 1-form symmetry group, $\alpha$ is an action of $G^{(0)}$ on $G^{(1)}$, and

$$[\Theta] \in H^3_\alpha(G^{(0)}, G^{(1)}), \tag{172}$$

in the cohomology twisted by the action $\alpha$.

An irreducible 1-charge under $\mathbb{G}^{(2)}$, furnished by an irreducible multiplet $M_L$ carrying a line $L$, can be described by the following pieces of information:

1. The induced 0-form symmetry $H_L \subseteq G^{(0)}$ on $L$ and the 1-form charge $\chi_L \in \widehat{G}^{(1)}$ of $L$, which are constrained such that

$$\chi_L \in \widehat{G}^{(1)}_{H_L} \subseteq \widehat{G}^{(1)} \,, \tag{173}$$

and

$$\chi_L\left([\Theta]_{H_L}\right) = 0 \in H^3(H_L, U(1)). \tag{174}$$

2. The fusion of topological junction local operators generating the induced $H_L$ 0-form symmetry on $L$ produces extra $\mathbb{C}^\times$ factors captured by a 2-cochain $\sigma$ on $H_L$ satisfying

$$\chi_L(\Theta_{H_L}) = \delta\sigma \,. \tag{175}$$

Rescalings of these local operators induces an equivalence relation

$$\sigma \sim \sigma + \delta\beta \,, \tag{176}$$

where $\beta$ is a $\mathbb{C}^\times$ valued 1-cochain on $H_L$. Thus only the class $[\sigma]$ of 2-cochains upto the above equivalence relation differentiates different 1-charges.

This is precisely the information describing irreducible 2-representations of the 2-group $\mathbb{G}^{(2)}$ [26, 66], thus fully justifying the $p = 2, q = 1$ piece of statement 1.4.

---

**Statement 6.1: 1-charges for 2-group symmetries**

1-charges of a $\mathbb{G}^{(2)}$ 2-group symmetry are 2-representations $\rho^{(2)}$ of the 2-group $\mathbb{G}^{(2)}$.

---

# 7 Conclusions and outlook

In this paper we answered the question, what the structure of charges for invertible generalized global symmetries is. The main insight that we gained is that these higher charges, or $q$-charges, fall into higher-representations of the symmetries.

This applies to standard 0-form symmetries (continuous and finite), but also higher-form symmetries and more generally higher-group symmetries. Thus, even when restricting one's attention to invertible symmetries, a higher-categorical structure emerges naturally. We have argued for the central relevance of higher-representations from a physical perspective – thus making their natural occurrence (and inevitability) apparent. The standard paradigms of extended $p$-dimensional operators being charged under $p$-form symmetries $G^{(p)}$, i.e. forming representations of these groups, are naturally obtained as specializations of the general structure presented here. The important insight is however, that this is by far only a small subset of generalized charges!

We discussed charged operators that are genuine and those that are non-genuine (e.g. operators appearing at the ends of higher-dimensional operators), including twisted sector operators. There is a natural higher-categorical structure that organizes these non-genuine charges.

We provided several examples in various spacetime dimensions ($d = 2, 3, 4$). However the full extent of higher-representations of invertible symmetries deserves continued in depth study. For instance, our examples of higher-charges of higher-form/group symmetries were focused on finite symmetries, but as we pointed out, the results should equally apply to continuous symmetries.

In view of the existence of non-invertible symmetries in $d \geq 3$, a natural question is to determine the higher-charges in such instances as well. This is the topic of Part II of this series [1]. Already here we can state the main tool to study these, which is the Symmetry TFT (SymTFT) [70–74] or more categorically, the Drinfeld center of the symmetry category.

# Acknowledgments

We thank Lea Bottini, Dan Freed, Theo Johnson-Freyd, David Jordan and Apoorv Tiwari for discussions. We also thank the KITP at UC Santa Barbara for hospitality.

**Funding information** This work is supported in part by the European Union's Horizon 2020 Framework through the ERC grants 682608 (LB, SSN) and 787185 (LB). SSN acknowledges support through the Simons Foundation Collaboration on "Special Holonomy in Geometry, Analysis, and Physics", Award ID: 724073, Schäfer-Nameki, and the EPSRC Open Fellowship EP/X01276X/1.

# A Notation and terminology

Let us collect some key notations and terminologies used in this paper.

- $\mathfrak{T}$: A QFT in $d$ spacetime dimension. Throughout the paper we assume that $\mathfrak{T}$ contains only a single topological local operator (up to multiplication by a $\mathbb{C}^\times$ element), namely the identity local operator.

- $d$: Spacetime dimension of the bulk QFT in discussion.

- Operator: A term used to refer to an arbitrary operator, which may or may not be topological. The term is used to refer to both genuine and non-genuine operators. Almost always we assume that the operator is simple: see section 2.2 for the definition.

- Genuine operator: A term used to refer to operators that are not attached to any higher-dimensional operators.

- Non-genuine operator: A term that is used when a typical operator in the class of operators being referred to is a non-genuine operator, i.e. is attached to higher-dimensional operators. However, special cases in that class of operators may include genuine operators.

- Defect: A term used interchangeably with the term 'operator'.

- $G^{(p)}$: A $p$-form symmetry group of $\mathfrak{T}$.

- $\mathbb{G}^{(p)}$: A $p$-group acting as symmetries of $\mathfrak{T}$.

- $D_p$: A $p$-dimensional *topological* operator.

- $\mathcal{O}_p$: A $p$-dimensional operator, which may or may not be topological.

- Untwisted sector operator: Another term used to refer to a genuine operator.

- Twisted sector operator: A term used to refer to a non-genuine operator arising at the boundary of a topological operator of one higher dimension.

- Local operator: An operator of dimension 0.

- Extended operator: An operator of dimension bigger than 0.

# B  Higher-representations

In this appendix, we introduce the mathematics of higher-representation theory for groups and higher-groups.

## B.1  Representations of groups

Let us begin with usual representations of a group $G^{(0)}$. Recall that a representation $\rho$ on a finite dimensional vector space $V$ is a map

$$\rho : \ G^{(0)} \to \text{End}(V), \tag{B.1}$$

where $\text{End}(V)$ is the set of endomorphisms of $V$, i.e. the set of linear maps from $V$ to itself. In order for it to be a representation, the map $\rho$ needs to satisfy the following additional conditions

$$\begin{aligned} \rho_g \rho_{g'} &= \rho_{gg'}, \\ \rho_1 &= 1, \end{aligned} \tag{B.2}$$

for all $g, g' \in G^{(0)}$.

Let us phrase the above in an alternative way, which opens up the generalization to higher-representations. Finite dimensional vector spaces form a linear category

$$\text{Vec}, \tag{B.3}$$

whose objects are vector spaces and morphisms are linear maps between vector spaces. On the other hand a group $G^{(0)}$ can be defined as a category $\mathcal{C}_{G^{(0)}}$, which has a single object and invertible morphisms. As there is a single object, all morphisms are endomorphisms of this object, which are identified with group elements $g \in G^{(0)}$. The composition of endomorphisms follows the group multiplication law.

A representation $\rho$ of $G^{(0)}$ can be specified equivalently as a functor

$$\rho : \mathcal{C}_{G^{(0)}} \to \mathsf{Vec}. \tag{B.4}$$

To see that this definition matches the usual one, note that the functor $\rho$ maps the single object of $\mathcal{C}_{G^{(0)}}$ to an object $V$ of $\mathsf{Vec}$, which is the underlying vector space for the representation $\rho$. Moreover, the functor $\rho$ maps the endomorphisms of the single object of $\mathcal{C}_{G^{(0)}}$ to endomorphisms of $V$.

## B.2 Higher-representations of groups

A $(q+1)$-representation $\rho^{(q+1)}$ of a group $G^{(0)}$ is defined similarly in terms of a functor

$$\rho^{(q+1)} : \mathcal{C}_{G^{(0)}}^{(q+1)} \to (q+1)\text{-Vec}, \tag{B.5}$$

between $(q+1)$-categories. Let us describe various ingredients appearing in the above equation below.

**Source $(q+1)$-category $\mathcal{C}_{G^{(0)}}^{(q+1)}$.** This is constructed as follows:

1. Start with the classifying space $BG^{(0)}$ of $G^{(0)}$ along with a marked point. The $(q+1)$-category $\mathcal{C}_{G^{(0)}}^{(q+1)}$ has a single simple object corresponding to the marked point.

2. The simple 1-morphisms of $\mathcal{C}_{G^{(0)}}^{(q+1)}$ are all endomorphisms of this single object and correspond to loops based at the marked point in $BG^{(0)}$.

3. The simple 2-morphisms of $\mathcal{C}_{G^{(0)}}^{(q+1)}$ are 2-dimensional homotopies between loops.

4. The simple 3-morphisms of $\mathcal{C}_{G^{(0)}}^{(q+1)}$ are 3-dimensional homotopies between 2-dimensional homotopies, and so on until we encounter $(q+1)$-morphisms.

Note that we could just replace $BG^{(0)}$ by any topological space $X$ construct in this way a $(q+1)$-category associated to it. In fact, in discussing $(q+1)$-representations of a $p$-group $\mathbb{G}^{(p)}$ we will need the category

$$\mathcal{C}_{\mathbb{G}^{(p)}}^{(q+1)}, \tag{B.6}$$

constructed in this fashion from the classifying space $B\mathbb{G}^{(p)}$ of the $p$-group $\mathbb{G}^{(p)}$.

Since the essential information of the classifying space $BG^{(0)}$ is in its first homotopy group, the essential information of the $(q+1)$-category $\mathcal{C}_{G^{(0)}}^{(q+1)}$ is in its 1-morphisms.

**Target $(q+1)$-category $(q+1)$-Vec.** This is essentially the "simplest" linear fusion $(q+1)$-category which is Karoubi/condensation complete[4] [62, 75]. More concretely,

1. Vec is the category of finite dimensional vector spaces.

2. 2-Vec is the 2-category of finite semi-simple categories.

3. $(q+1)$-Vec for $q \geq 2$ is the $(q+1)$-category of fusion $(q-1)$-categories.

---

[4]According to the definition of [75], Karoubi completeness is part of definition of a fusion higher-category, so mentioning it again is redundant. But it is important to emphasize this point as one can obtain simpler linear $(q+1)$-categories with monoidal and other structures for $q \geq 2$ except that they are not Karoubi complete.

**$(q+1)$-Category that is simpler than $(q+1)$-Vec.** It should be noted that, for $q \geq 2$, there is a sub-$(q+1)$-category of $(q+1)$-Vec which we denote as

$$(q+1)\text{-Vec}^0,\qquad(\text{B.7})$$

which would be termed as the simplest "fusion" $(q+1)$-category if we drop the condition of Karoubi completeness. This higher-category $(q+1)$-Vec$^0$ has the following data:

1. The only simple object $(q+1)$-Vec$^0$ is the identity object **1** of $(q+1)$-Vec.

2. For a $(r+1)$-category $\mathcal{C}$, we define $\Omega(\mathcal{C})$ to be the $r$-category formed by the endomorphisms of the identity object of $\mathcal{C}$. Then, the only simple object of $\Omega^s\left((q+1)\text{-Vec}^0\right)$ is the identity object of $\Omega^s\left((q+1)\text{-Vec}\right)$, for all $1 \leq s \leq q+1$.

If one only works with $(q+1)$-Vec$^0$ then one never sees the important physical phenomenon of symmetry fractionalization discussed in the main text. In particular, one only observes group cohomology type $q$-charges/$(q+1)$-representations for a 0-form symmetry group $G^{(0)}$. In order to see non-group cohomology type $q$-charges/$(q+1)$-representations displaying symmetry fractionalization of 0-form symmetry group $G^{(0)}$, one has to pass to the full fusion $(q+1)$-category $(q+1)$-Vec.

Some papers in recent literature only work with $(q+1)$-Vec$^0$ while discussing $(q+1)$-representations. We emphasize that this only captures a small subset of $(q+1)$-representations if $q \geq 2$, and in fact a generic $(q+1)$-representation is not of this type.

## B.3 Higher-representations of higher-groups

As was already briefly remarked above, a $(q+1)$-representation $\rho^{(q+1)}$ of a $p$-group $\mathbb{G}^{(p)}$ is defined as a functor

$$\rho^{(q+1)}:\ \mathcal{C}^{(q+1)}_{\mathbb{G}^{(p)}} \to (q+1)\text{-Vec},\qquad(\text{B.8})$$

where the target $(q+1)$-category is the same as for $(q+1)$-representation of a group $G^{(0)}$ but the source $(q+1)$-category is now built from the classifying space $B\mathbb{G}^{(p)}$ of the $p$-group $\mathbb{G}^{(p)}$ as discussed above. In particular, a general $p$-group has $r$-form symmetry groups $G^{(r)}$ for $0 \leq r \leq p-1$, and we have

$$\left\{\text{Simple objects of } \Omega^{r+1}\left(\mathcal{C}^{(q+1)}_{\mathbb{G}^{(p)}}\right) \text{ upto isomorphism}\right\} = \left\{\text{Elements of } G^{(r)}\right\}.\qquad(\text{B.9})$$

Let us note that the $(d-1)$-category (11) can be obtained as the linearization of $\Omega(\mathcal{C}^{(d)}_{\mathbb{G}^{(p)}})$.

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
