# Peer review of "Generalized Charges, Part I: Invertible Symmetries and Higher Representations"

_SciPost Physics, doi:SciPost Phys. 16, 093 (2024)_

## Round 3 · Referee Report · Anonymous (Referee 1) · 2023-12-27

Report

The manuscript presents a study of the representation theory that is appropriate for extended operators in QFTs transforming under ordinary, higher-form and higher-group symmetries. The main insight that is put forward is that in general, q-dimensional extended operators form a (q+1)-representation of the symmetry group (or higher-group), which is a higher-categorical generalization of an ordinary representation. In referee’s opinion, this is a result of fundamental importance, for which a systematic analysis provided by the current manuscript has been missing in the literature.

Throughout the paper, physical and convincing arguments based on the properties of topological symmetry defects and junctions, as well as their interplay with the charged extended operators, are provided, which strongly support the claim. For special values of q and the form degree of the symmetry, it is shown that the explicit mathematical data characterizing a (q+1)-representation naturally arise when one considers various configurations of symmetry defects and charged operators. The discussion also extends to include non-genuine extended operators as well as those in the twisted sector.

Overall, the manuscript includes a fundamental insight, and teaches us a natural place in QFT where higher-representation theory appears. It will likely be used as a future reference for further studies in the subject. Therefore, the referee recommends publication of the manuscript in SciPost.

Requested changes

Please find below minor comments and suggestions:

1) In Sections 2.3.2 and 2.3.3, the phenomenon of symmetry fractionalization is discussed. Although the presentation there is clear and interesting, it appears that it will be helpful for the readers if the relation between the current discussion and the existing literature on symmetry fractionalization is spelled out more explicitly.

Specifically, in the literature it is commonly said that the fractionalization of a 0-form symmetry is a phenomenon which can occur in the presence of other higher-form symmetries. Intuitively, this is understood as either dressing the junctions of 0-form symmetry defects with higher-form symmetry defects, or dually, as activating the background gauge field for the higher-form symmetries using that for the 0-form symmetry. For instance, in the presence of a 0-form symmetry G and a 1-form symmetry A (with vanishing Postnikov class), it is known that the possible symmetry fractionalizations are in 1-to-1 correspondence with the elements of H^2(G,A).

The discussions in sections 2.3.2 and 2.3.3 appear to be different from such a phenomenon, since the 0-form symmetry there seems to fractionalize in the absence of any higher-form symmetries. It will be very helpful if this point is further clarified.

2) In Section 2.3.3, a novel possibility where an invertible 0-form symmetry fractionalizes to a “localized” non-invertible symmetry supported on the worldvolume of an extended operator. In Example 2.6, although it is convincing that such a structure mathematically exists as a 3-representation of the group Z_2, it will perhaps greatly strengthen the proposal of the manuscript (in particular Statement 1.2) if an actual QFT example where this occurs is mentioned.

Below are some typos:

  • On page 8, for the 3rd of the itemized list, subscript of "O_q^(a,b;B)" must be q-1.

  • On page 25, "factionalizing" must be "fractionalizing".

  • On page 71, "moreover" is not capitalized.

---

## Round 3 · Referee Report · Anonymous (Referee 2) · 2024-1-13

Report

This paper introduced the concept of q-charge, which generalizes the notion of charge for 0-form symmetries. The author argued that in analogous to the relation between charges and representations for 0-form symmetries, q-charges are related to the (q+1)-representations of the corresponding (q+1)-groups. The author also discussed the q-charges furnished by non-genuine operators (operators attached with higher-dimensional operators) and proposed a correspondence between higher q-charges and higher (q+1)-categories.

This paper is written with very good clarity, offering clear definitions and statements, accompanied by examples that support the statements. While this paper did not offer significant innovations, it places many known elements into a coherent framework. I recommend the publication of this paper on SciPost.

---

## Round 4 · Author Response

We thank both referees for their positive reviews. We have addressed the points raised by referee 1 as described in "List of changes". We have also corrected the typos pointed out by referee 1. With these modifications, we hope that the paper is now suitable for publication in SciPost.

---

## Round 4 · List of Changes

1) This is an important distinction that we have clarified in footnote 3 on page 22.
2) We do not know of an example of a defect exhibiting this 2-charge, but we have provided a suggestion in Example 2.6 on how such a 2-charge may be produced.

---

## Editorial Decision

published